# Stable Fair Graph Representation Learning with Lipschitz Constraint

Qiang Chen [* 1]   Zhongze Wu [* 1]   Xiu Su [† 1]   Xi Lin [† 2]   Zhe Qu [1]   Shan You [3]   Shuo Yang [4]   Chang Xu [5]

## Abstract

Group fairness based on adversarial training has gained significant attention on graph data, which was implemented by masking sensitive attributes to generate fair feature views. However, existing models suffer from training instability due to uncertainty of the generated masks and the trade-off between fairness and utility. In this work, we propose a stable fair Graph Neural Network (SFG) to maintain training stability while preserving accuracy and fairness performance. Specifically, we first theoretically derive a tight upper Lipschitz bound to control the stability of existing adversarial-based models and employ a stochastic projected subgradient algorithm to constrain the bound, which operates in a block-coordinate manner. Additionally, we construct the uncertainty set to train the model, which can prevent unstable training by dropping some overfitting nodes caused by chasing fairness. Extensive experiments conducted on three real-world datasets demonstrate that SFG is stable and outperforms other state-of-the-art adversarial-based methods in terms of both fairness and utility performance. Codes are available at https://github.com/sh-qiangchen/SFG.

## 1. Introduction

Graph Neural Networks (GNNs)(Kipf & Welling, 2017; Hamilton & Ying, 2017) have recently emerged as a crucial tool for modeling and representing graph-structured data, which are widely used in various applications such as recommendation systems(Gao et al., 2022) and drug discovery(Rozemberczki et al., 2022). Despite significant success,

---
[*]Equal contribution   [1]Central South University, Changsha, Hunan, China [2]Shanghai Jiaotong University, Shanghai, China [3]SenseTime Research, Shanghai, China [4]Harbin Institute of Technology (Shenzhen), Shenzhen, China [5]University of Sydney, Sydney, Australia. Correspondence to: Xiu Su <xiusu1994@csu.edu.cn>, Xi Lin <linxi234@sjtu.edu.cn>.

*Proceedings of the 42ⁿᵈ International Conference on Machine Learning*, Vancouver, Canada. PMLR 267, 2025. Copyright 2025 by the author(s).

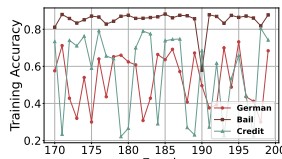 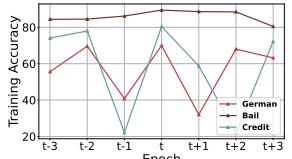

(a) Training accuracy curve    (b) Accuracy near the optimum

*Figure 1.* Instability of the existing adversarial-based model utility on German/Bail/Credit dataset. (a) visualizes the accuracy curve in the convergence phase. (b) visualizes the accuracy near the optimal model.

GNNs can suffer from fairness issues that can be divided mainly into individual fairness(Dong et al., 2021) and group fairness(Dai & Wang, 2021). Group fairness highlights mitigating bias in the demographic groups defined by sensitive attributes, *e.g.*, gender, race(Agarwal et al., 2021). In other words, group fairness concentrates on ensuring that the outputs of GNNs are independent of the sensitive attribute.

Recent studies(Dai & Wang, 2021; Wang et al., 2022; Zhu et al., 2024; Li et al., 2024) have revealed that the removal of sensitive attribute-related information can force GNNs to make decisions independently of the sensitive attribute. The core idea behind most of these approaches is masking sensitive-relevant features by a learnable mask and adopting adversarial learning(Ling et al., 2023) to learn fair node representations. However, in our empirical investigations, we observed that when employing learnable masks and adversarial-based methods to optimize the framework for both model fairness and utility on real datasets, the training process suffers from significant instability. This instability results in the selection of an optimal model becoming largely accidental and not trustworthy(Zhang et al., 2024; Jing et al., 2024). As shown in Figure 1, we see that the accuracy curve in the convergence phase is still very unstable, and the accuracy near the optimal model fluctuates greatly.

To ensure the reliability of the model in the trade-off between model utility and fairness, it is urgently needed to maintain the stability of the training process. Most existing methods(Scaman & Virmaux, 2018; Zhao et al., 2021; Agarwal et al., 2021; Jia et al., 2024) maintain stability by constraining the Lipschitz bound of the model, which is a separable accumulation form of each component parameter, or using the Jacobian matrix to approximate. These forms

are loose and need a tight Lipschitz bound. In the context of GNNs, the Lipschitz constraint is mainly used to control the stability of the GNN encoder(Jia et al., 2023; Juvina et al., 2024; Su et al., 2024), and is not suitable when there are other modules, such as generator in adversarial-based methods.

In this work, we propose Stable Fair GNN(SFG), a stable graph representation learning framework for preserving fairness while maintaining training stability. We first theoretically derive an easy-to-compute non-separable upper Lipschitz bound based on GNN message passing mechanism and graph spectral theory for fair graph representation learning. The derived bound can control the overall stability of the GNN encoder and mask generator of adversarial-based methods, and it's tight compared to separable bound.

Motivated by our theoretical insights, we use a stochastic projected subgradient strategy that operates in a block-coordinate manner to constrain the weights of adversarial-based fair graph model. This method can convert non-covex problems into multi-convex ones. Additionally, we construct the uncertainty set within the chi-squared divergence ball surrounding the generated fair view distribution to enhance encoder robustness, which can prevent unstable training of the encoder by dropping some overfitting nodes caused by chasing fairness. Our contributions are as follows:

- To the best of our knowledge, this is the first attempt to study the stability of graph fairness learning with generator. We theoretically derive an easy-to-compute tight upper Lipschitz bound to control the stability of the framework, which is suitable for fair graph model with generator. Based on theoretical insights, we use the block coordinate approach to operate layerwise.

- To further enhance robustness, we construct the uncertainty set within the chi-squared divergence ball surrounding the generated fair view distribution to avoid overfitting caused by chasing fairness and instability caused by distribution shifts due to generated mask.

- We conducted extensive experiments on three real-world datasets, demonstrating the superior stability of SFG over the existing adversarial-based model FaiVGNN, which outperformed other state-of-the-art fairness models in fairness and accuracy performance.

## 2. Preliminary

Let $G = (V, E, X)$ denote an undirected attributed graph, comprised of a set of $|V| = K$ nodes $V = \{v_1, ..., v_K\}$ and a set of $|E|$ edges $E$. $X \in \mathbb{R}^{K \times N_0}$ represents the nodes features matrix, where $N_0$ is the dimension of the node feature. $A \in \{0, 1\}^{K \times K}$ is the adjacency matrix where $A_{uv} = 1$ indicates that there is an edge $e_{uv} \in E$

between the node $u$ and $v$, and $A_{uv} = 0$ otherwise. For every node $v$, we define its neighborhood $N(v)$ as the set of nodes $u$ such that there exists an edge going from $u$ to $v$. The goal of graph node classification is to learn a representation vector $h_v$ of $v$ such that $v$'s label can be predicted as $y_v = f(h_v) \in \{1, ..., C\}$. In this work, we focus on the node classification task while learning fair node representations.

**Graph Neural Network.** Modern GNNs follow the message passing mechanism (Gilmer et al., 2017), which iteratively updates the representation of a node by aggregating representations of its neighbors. Formally, node $v \in V$ at the $i$-th layer of a GNN is updated by:

$$h_v^{(i)} = ReLu(w_0^{(i)} h_v^{(i-1)} + w_1^{(i)} \sum_{u \in N(v)} \rho_{v,u} h_u^{(i-1)}), \quad (1)$$

where $h_v^{(i)} \in \mathbb{R}^{N_i}$ is the embedding vector of node $v$ at the $i$-th layer, $w_0^{(i)}$ and $w_1^{(i)}$ are weight matrices in $\mathbb{R}^{N_i \times N_{i-1}}$. The activation function for all layers is Relu, which is 1-Lipschitz, except for the last layer. Equation (1) can be expressed under the vector form:

$$H^{(i)} = ReLu(W_0^{(i)} H^{(i-1)} + W_1^{(i)} H^{(i-1)}), \quad (2)$$

where $H^{(i)} \in \mathbb{R}^{KN_i}$, $W_0^{(i)}$ and $W_1^{(i)}$ are matrices with dimensions $KN_i \times KN_{i-1}$, and the total amount of parameters remains the same, except that they are tiled $K \times K$ times. Thus, we can have follow expression:

$$W_0^{(i)} = Id_K \otimes w_0^{(i)}, W_1^{(i)} = M \otimes w_1^{(i)}, \quad (3)$$

where $\otimes$ denotes the standard matrix Kronecker product. It is easy to see that $M$ is a graph shift operator(Gama et al., 2020; Jing et al., 2023), such as a normalized Laplacians matrix, and it is closely related to stable representations.

**Lipschitz Constant.** A function $f : \mathbb{R}^n \to \mathbb{R}^m$ is said to be Lipschitz continuous on an input set $\mathcal{X} \subseteq \mathbb{R}^n$ if there exists a bound $L \geq 0$ such that for all $\delta, x \in \mathcal{X}$, and $\delta$ is a small perturbation of $x$, $f$ satisfies:

$$\| f(x + \delta) - f(x) \| \leq L \| (x + \delta) - x \|, \quad (4)$$

where $\| \cdot \|$ is the distance norm and $L$ is a Lipschitz constant of the function, denoted as $Lip(f)$. The Lipschitz constant essentially quantifies the maximum change in the output of a function corresponding to a small perturbation in its input. In adversarial-based fair models, softly masking node features is equivalent to imposing a perturbation on the features, so the difference of masks can be viewed as $\delta$. Due to the fact that an exact Lipschitz bound is computationally expensive(Virmaux & Scaman, 2018), recent works(Szegedy et al., 2014; Juvina et al., 2024; Jia et al., 2024) propose some approximate bounds. However, these bounds are loose and only suitable for the GNN encoder.

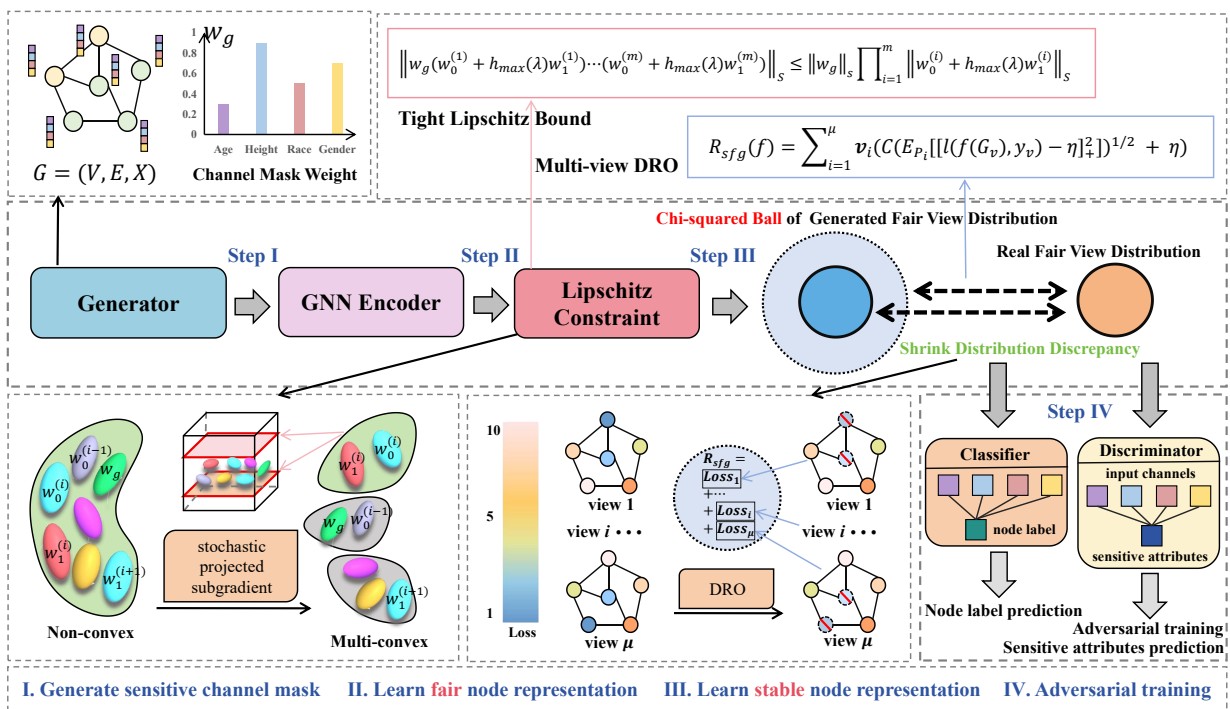

Figure 2. The framework of the Stable Fair Graph Neural Net Work(SFG). Compared to FairVGNN(Wang et al., 2022), our model uses two methods to maintain stability. (1) A tight Lipschitz bound is derived to control fluctuations in the representation learned by the encoder and generator. (2) DRO makes the encoder robust to different fair views.

## 3. Method

In Section 3.1, we theoretically derive a tight upper Lipschitz bound to control the stability of a fair graph model. In Section 3.2, we employ a stochastic projected subgradient algorithm to control the Lipschitz bound in the presence of a mask generator, which operates in a block-coordinate manner. In Section 3.3, we propose to train the worst-case distribution within the chi-squared divergence ball surrounding the generated fair view distribution to further enhance stability.

### 3.1. Estimating Lipschitz Constant

To analyze the stability of the output of the fair graph model, we check the details of these models. As shown in Figure 2, these models typically use a generator to create a sensitive channel mask, an encoder for learning fair node representation, a discriminator for sensitive attribute prediction and a classifier for node label prediction. All components above consistently employ the Binary Cross-Entropy (BCE) loss function(Su et al., 2021c;d) for optimization. For detailed implementation of these fundamental components in this work, refer to Appendix D.5.

The uncertainty of generator and the trade-off between model fairness and utility may lead learned the embeddings

of nodes to fluctuate dramatically. Obviously, controlling Lipschitz bound is a promising direction to keep the model training process stable. Compared to weight climbs of FairVGNN, Lipschitz bound constraint does not lead to suboptimal solutions. Truncating large weights element-wise can weaken the influence of features with significant weights and alter the update direction of parameters(Su et al., 2021b), which causes the optimization path to deviate from the true direction guided by the gradients, and these would lead to suboptimal solutions.

**Theorem 3.1.** *Let $Y$ be the output of a mask generator and an m-layer GNN encoder (denoted as $f(\cdot)$) with $X$ as input. Assuming that the matrix $M$ in Equation (3) is symmetric with nonnegative elements and the weights of the mask generator and GNN encoder are also nonnegative, and the activation function (represented in $\rho(\cdot)$) is ReLU with a Lipschitz bound of $Lip(\rho) = 1$, then the cumulative Lipschitz bound of the mask generator and the GNN encoder, $Lip(f)$, satisfies the following:*

$$
\begin{aligned}
Lip(f) =\parallel (w_0^{(m)} + h_{max}(\lambda)w_1^{(m)}) \\
\ldots (w_0^{(1)} + h_{max}(\lambda)w_1^{(1)}) \odot w_g \parallel_S ,
\end{aligned} \tag{5}
$$

*where $\parallel \cdot \parallel_S$ denotes the spectral norm, $\odot$ represents element-wise product supporting broadcast, $w_g$ denotes the weights of mask generator, $w_0^{(i)}$ and $w_1^{(i)}(i \in 1, \ldots, m)$*

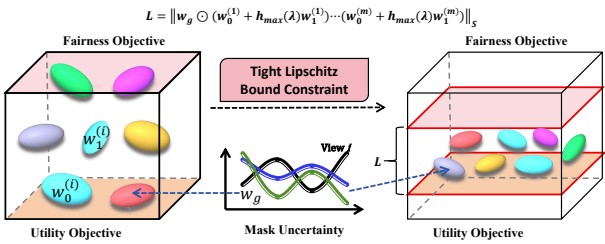

$$L = \left\| w_g \odot (w_0^{(1)} + h_{max}(\lambda)w_1^{(1)}) \cdots (w_0^{(m)} + h_{max}(\lambda)w_1^{(m)}) \right\|_S$$

*Figure 3.* Comparison of weights variation space.

*are the weights of the GNN encoder, and $h_{max}(\lambda)$ is the maximal frequency response of the graph filter from graph spectral theory perspectives, and $\lambda$ is the eigenvalue of symmetric normalized Laplacians matrix.*

*Proof Sketch*: The weights of generator and encoder at layer $i$ are represented as $w_g \in \mathbb{R}^{1 \times N_0}$ and $W^{(i)} = W_0^{(i)} + W_1^{(i)}$ respectively. $W^{(i)} = (U \otimes Id_{N_i})(Id_K \otimes w_0^{(i)} + h(\Lambda) \otimes w_1^{(i)})(U^T \otimes Id_{N_{i-1}})$, which is obtained on the basis of the distributive and associative laws of the Kronecker products. Since the dimensions of generator's weights do not match encoder's, we cannot multiply them directly. We first form it into a diagonal matrix $W_{g_1} \in \mathbb{R}^{N_0 \times N_0}$. Then we tile it along the diagonal $K$ times and get a matrix $W_g \in \mathbb{R}^{KN_0 \times KN_0}$, which is also a diagonal matrix. This is the most critical step. Finally, we get the following expression:

$$
\begin{aligned}
&W^{(m)} \dots W^{(1)} W_g \\
&= (U \otimes Id_{N_m})(Id_K \otimes w_0^{(m)} + h(\Lambda) \otimes w_1^{(m)}) \\
&\dots (Id_K \otimes w_0^{(1)} + h(\Lambda) \otimes w_1^{(1)})(U^T \otimes Id_{N_0}) W_g \quad (6) \\
&= (U \otimes Id_{N_m})(Id_K \otimes w_0^{(m)} + h(\Lambda) \otimes w_1^{(m)}) \\
&\dots (Id_K \otimes w_0^{(1)} + h(\Lambda) \otimes w_1^{(1)}) W_g(U^T \otimes Id_{N_0}).
\end{aligned}
$$

Equation (6) showns that $k$-th block diagonal matrix of eigenvalue matrix of $W^{(m)} \dots W^{(1)} W_g$ is $(w_0^{(m)} + h_k(\lambda)w_1^{(m)}) \dots (w_0^{(1)} + h_k(\lambda)w_1^{(1)})W_{g_1}$. So we get our result:

$$
\begin{aligned}
Lip(f) &= \| W^{(m)} \dots W^{(1)} W_g \|_S \\
&= \| (w_0^{(m)} + h_{max}(\lambda)w_1^{(m)}) \\
&\dots (w_0^{(1)} + h_{max}(\lambda)w_1^{(1)} W_{g_1}) \|_S \\
&= \| (w_0^{(m)} + h_{max}(\lambda)w_1^{(m)}) \\
&\dots (w_0^{(1)} + h_{max}(\lambda)w_1^{(1)}) \odot w_g \|_S \\
&\leq \| w_g \|_S \prod_{i=1}^{m} \| w_0^{(i)} + h_{max}(\lambda)w_1^{(i)} \|_S .
\end{aligned}
\quad (7)
$$

The result shows the Lipschitz bound of the encoder with the participation of the mask generator weight. Compared

with just restricting the encoder, SFG can comprehensively consider the uncertainty of the generated mask so that the encoder can be robust to different masks. Additionally, our non-separable upper Lipschitz bound is tighter than the separable counterpart. Figure 3 illustrates how the Lipschitz bound controls the size of the weight variation space, thereby enabling the learning of stable representations.

For detailed proof processes, please refer to Appendix B.2. According to Theorem 3.1, for the low-pass graph filter, $h(\lambda) = 1 - \lambda \in [-1, 1]$ due to the eigenvalue of symmetric normalized Laplacians matrices $\lambda \in [0, 2]$, which corresponds to $M = D^{-1/2}AD^{-1/2}$, it is actually Graph-SAGE graph convolution operator using the mean aggregator function. For the high-pass graph filter, $h(\lambda) \in [0, 2]$ and $w_0^{(i)} = 0$ for all $i \in 1, \dots, m$. For a band-pass filter like BernNet(He et al., 2021), $h(\lambda) \in [0, 1]$ and $w_0^{(i)} = 0$. Another interesting point is that this approach can somewhat bridge the gap between GNN spatial and spectral domains.

### 3.2. Controlling Lipschitz Constant

To control the stability of the output of the fair graph model, we consider maintaining the stability of the GNN encoder with the participation of the mask generator. Specifically, we use a gradient descent algorithm to optimize the generator parameters $w_g$ and the GNN encoder parameters $w_0(i)$ and $w_1(i)$ for all $i$. Then after each epoch, consider controlling the Lipschitz bounds of the GNN encoder in the presence of the generated mask. In this work, we study the most widely used GNN, namely GraphSAGE, it satisfies:

$$\| (w_0^{(m)} + w_1^{(m)}) \dots (w_0^{(1)} + w_1^{(1)}) \odot w_g \|_S \leq \tau , \quad (8)$$

where $\tau$ is manually-set thresholds.

The bound on the product in Equation (8) is a non-convex function, we use a projected stochastic subgradient algorithm to optimize the weights, which converts a non-convex problem into a multi-convex problem. The detailed process of parameter constraint is described as follows. At epoch $t > 0$, we use the layer-wise operating block coordinate approach to control the weights, and denote $w_0^{(i,t)}$ and $w_1^{(i,t)}$ as the unconstrained weight matrices for layer $i$ at epoch $t$, and $w_g^{(t)}$ are the mask generator weights and define $W^{(i,t)} = [w_0^{(i,t)}, w_1^{(i,t)}]^T$. Therefore, Equation (8) can re-express as:

$$\| A_{old}^{(i,t)} W^{(i,t)} B_{new}^{(i,t)} \|_S \leq \tau, \quad (9)$$

where matrices $A_{old}^{(i,t)}$ and $B_{new}^{(i,t)}$ represent the product of the cumulative weights for the next and previous layers,

**Algorithm 1** The accelerated DFB algorithm with mask generator

---

**Input:** $W^{(i,t)}, A_{old}^{(i,t)}, B_{new}^{(i,t)}, Y_0 \in \mathbb{R}^{N_m \times N_0}$
**Initialize:** $\gamma = 1/(\| A_{old}^{(i,t)} \|_s \| B_{new}^{(i,t)} \|_s)^2, \alpha \in [2, \infty)$
**for** $i = 1$ **to** $nit$ **do**
    $\zeta = 1/(i + 1 + \alpha)$
    $Z_l = Y_l + \zeta(Y_l - Y_{l-1})$
    $V_l = proj_{\mathcal{D}}(W^{(i,t)} - A_{old}^{(i,t)^T} Z_l B_{new}^{(i,t)^T})$
    $\hat{Y}_l = Z_l + \gamma A_{old}^{(i,t)^T} V_l B_{new}^{(i,t)^T}$
    $Y_{l+1} = \hat{Y}_l - \gamma proj_{\mathcal{B}(0,\tau)}(\gamma^{-1}\hat{Y}_l)$
**end for**
**Return:** $V_l$

---

respectively:

$$B_{new}^{(i,t)} = [I_{N_{i-1}}, I_{N_{i-1}}]W^{(i-1,t+1)} \dots [I_{N_0}, I_{N_0}] \odot w_g^{(t)},$$
$$A_{old}^{(i,t)} = [I_{N_m}, I_{N_m}]W^{(m,t)} \dots W^{(i+1,t)}[I_{N_i}, I_{N_i}]. \quad (10)$$

Accordingly, for the first layer with $i = 1$, we have $B_{new}^{(0,t)} = [I_{N_0}, I_{N_0}] \odot w_g^{(t)}$.

The weights set is a closed convex set based on Equation (9) and the non-negativity assumption. We utilize an instance of the accelerated iterative dual forward-backward (DFB) algorithm, as proposed by (Neacşu et al., 2024), to update $W^{(i,t)}$. The projection algorithm is detailed in Algorithm 1.

Here, $proj_{\mathcal{D}}$ represents the projection onto the cone of non-negative weights, whereas $proj_{\mathcal{B}(0,\tau)}$ represents the projection onto the spectral ball of radius $\tau > 0$.

### 3.3. Fair View Distributional Robust Optimization

To enhance the model's adaptability to mask variations, we employ Distributionally Robust Optimization (DRO) to bolster the robustness of the encoder. We use $\boldsymbol{G_v}$ as a random variable of node $v$ ego-graphs whose realization is $Gv = (A_v, E_v, X_v) \subseteq G$, and a whole graph can be fragmented as a set of instances $(G_v, y_v)_{v \in V}$. Therefore, the conditional distribution $P(\boldsymbol{Y}|\boldsymbol{G})$ can be decomposed as a product of $|V|$ independent and identical marginal distributions $P(\boldsymbol{y}|\boldsymbol{G_v})$. Typically, adversarial-based fair graph models employ ERM to find a optimal model $f$ that minimizes the average risk under the training distribution $P(\boldsymbol{Y}|\boldsymbol{G})$:

$$\min_{f \in \mathcal{F}}\{\mathcal{R}_{erm}(f) \coloneqq \mathbb{E}_{(G_v,y_v) \in P(\boldsymbol{Y}|\boldsymbol{G})}[l(f(G_v), y_v)]\}. \quad (11)$$

The use of mask generator in the fair graph model will cause the distribution shifts for $P(\boldsymbol{y}|\boldsymbol{G_v})$, i.e., different fair feature view will be used as input to the GNN encoder. However, models trained through ERM are highly based on spurious correlations that do not always hold under distributional shifts(Arjovsky et al., 2019), which would cause

training instability. To further enhance the stability of the fair graph model, we use DRO minimizes the worst-case risk over an uncertainty set $Q$ by solving:

$$\min_{f \in \mathcal{F}}\{\mathcal{R}_{dro}(f) \coloneqq \sup_{Q \in \mathcal{P}(P)} \mathbb{E}_{(G_v,y_v) \in Q}[l(f(G_v), y_v)]\}. \quad (12)$$

Here the uncertainty set $Q$ approximates potential fair feature view distributions, which is usually formulated as a divergence ball with a radius of $r$ surrounding the training distribution $\mathcal{P}(P) = \{Q : D(Q, P(\boldsymbol{Y}|\boldsymbol{G})) \le r\}$ limited with a distance metric $D(\cdot, \cdot)$ such as f-divergence(Namkoong & Duchi, 2016) or Wasserstein distance(Abadeh et al., 2018). Unless otherwise specified, $P$ or $P(\boldsymbol{Y}|\boldsymbol{G})$ denotes the distribution of a generated view. In this work, we use $\chi^2$-divergence as instance of $D(\cdot, \cdot)$, which given by:

$$D_{\chi^2}(Q||P) = \int (\frac{dQ}{dP} - 1)^2 dP. \quad (13)$$

**Lemma 3.2** (See (Duchi & Namkoong, 2021)). *Let $\alpha_{min}$ be the ratio between the size of the smallest domain and the size of the training data,$\mathcal{R}_{dro}(f)$ is equal to the following expression:*

$$\inf_{\eta \in \mathbb{R}}\{F(f,\eta) \coloneqq C(\mathbb{E}_P[[l(f(G_v), y_v) - \eta]_+^2])^{\frac{1}{2}} + \eta\}, \quad (14)$$

*where $C = (2(1/\alpha_{min} - 1)^2 + 1)^{1/2}$ .*

Denoting by $\eta^*$ the optimal dual variable in Equation (14), it is easy to see that large losses above $\eta^*$ are upweighted due to the squared term.

**Lemma 3.3** (See (Zhai et al., 2021)). *Let $\mathcal{R}_{max}(f)$ denotes the worst-case risk on $P$, then we have:*

$$\mathcal{R}_{max}(f) \le \mathcal{R}_{dro}(f) \le F(f, \eta). \quad (15)$$

**Proposition 3.4.** *Let the distribution of the fair feature view generated be denoted by $P_i, i \in 1, \dots, \mu$. Minimizing the worst-case risk on all views, is equal to minimize the following expression:*

$$\mathcal{R}_{sfg}(f) \coloneqq \sum_{i=1}^{\mu} \boldsymbol{v}_i(C(\mathbb{E}_{P_i}[[l(f(G_v), y_v) - \eta]_+^2])^{1/2} + \eta), \quad (16)$$

*where $C = (2(1/\alpha_{min} - 1)^2 + 1)^{1/2}$, and $\alpha_{min}$ is the smallest size of the entire group among all views. $\boldsymbol{v} \in \Delta_\mu$, $\Delta_\mu$ is a $(\mu - 1)$-dimensional probability simplex, and $\boldsymbol{v}_i$ represents distance between the true mask and $i$-th generated mask.*

For detailed proof processes, please refer to Appendix C. We can see from Proposition 3.4 that our proposed robust model actually ignores some views and the nodes that cause overfitting during fairness training. This overfitting would cause the accuracy performance to deteriorate dramatically, leading to an unstable graph fair model. For simplicity, we choose uniform $\boldsymbol{v}$ in our work, and we use Brent's method(Brent, 1971) to find the $\eta^*$.

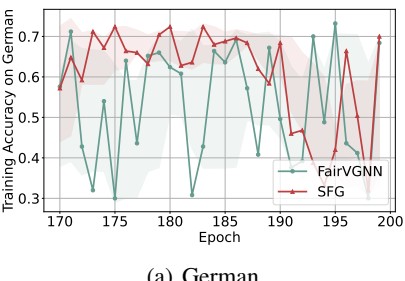

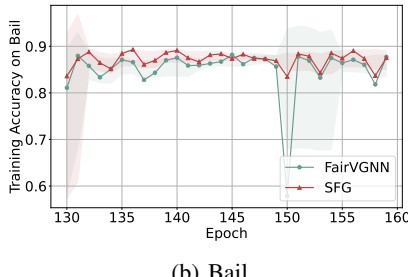

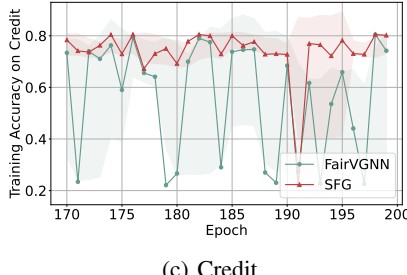

| (a) German | (b) Bail | (c) Credit |

*Figure 4.* Training accuracy curve on German, Bail and Credit datasets. The shaded area represents the rolling standard deviation with a window size of 5, which can be used to measure the stability between epochs.

*Table 1.* Comparison results of SFG with baseline fairness methods. In each row, the best result is indicated in bold, while the runner-up result is marked with an underline.

| Dataset | Metric | GCN | Nifty | FairGNN | FairVGNN | FairSAD | SFG |
|---|---|---|---|---|---|---|---|
| German | Acc | $68.72 \pm 2.38$ | $69.92 \pm 0.81$ | $\mathbf{71.32 \pm 0.16}$ | $69.84 \pm 0.74$ | $68.29 \pm 2.80$ | $\underline{70.32 \pm 0.46}$ |
| | AUC | $67.64 \pm 6.49$ | $72.05 \pm 2.15$ | $69.30 \pm 8.60$ | $66.74 \pm 4.08$ | $\mathbf{72.20 \pm 3.42}$ | $\underline{69.38 \pm 4.77}$ |
| | $\Delta_{DP}$ | $13.91 \pm 12.24$ | $4.21 \pm 2.51$ | $7.12 \pm 8.25$ | $2.54 \pm 1.75$ | $\mathbf{0.00 \pm 0.00}$ | $\underline{1.36 \pm 2.61}$ |
| | $\Delta_{EO}$ | $11.01 \pm 11.16$ | $2.66 \pm 2.17$ | $5.67 \pm 5.16$ | $3.06 \pm 1.50$ | $\mathbf{0.00 \pm 0.00}$ | $\underline{0.54 \pm 0.74}$ |
| Bail | Acc | $82.01 \pm 0.24$ | $77.38 \pm 1.65$ | $83.59 \pm 0.86$ | $\underline{89.24 \pm 0.49}$ | $83.73 \pm 1.22$ | $\mathbf{89.49 \pm 1.74}$ |
| | AUC | $85.91 \pm 0.12$ | $80.43 \pm 0.39$ | $87.55 \pm 0.60$ | $\underline{91.99 \pm 0.34}$ | $88.46 \pm 0.81$ | $\mathbf{93.98 \pm 1.41}$ |
| | $\Delta_{DP}$ | $9.21 \pm 0.16$ | $5.74 \pm 0.38$ | $6.51 \pm 0.76$ | $4.39 \pm 0.96$ | $\mathbf{2.08 \pm 1.24}$ | $\underline{3.55 \pm 0.58}$ |
| | $\Delta_{EO}$ | $6.29 \pm 0.04$ | $4.07 \pm 1.28$ | $4.50 \pm 1.09$ | $2.49 \pm 1.06$ | $\underline{2.14 \pm 1.18}$ | $\mathbf{1.11 \pm 0.43}$ |
| Credit | Acc | $73.67 \pm 0.03$ | $69.63 \pm 0.75$ | $76.16 \pm 1.82$ | $\underline{80.13 \pm 0.32}$ | $77.41 \pm 0.78$ | $\mathbf{80.34 \pm 0.11}$ |
| | AUC | $73.87 \pm 0.02$ | $69.08 \pm 0.15$ | $74.62 \pm 0.87$ | $\underline{74.17 \pm 0.54}$ | $71.72 \pm 0.51$ | $\mathbf{74.37 \pm 0.37}$ |
| | $\Delta_{DP}$ | $12.86 \pm 0.09$ | $10.34 \pm 1.15$ | $6.87 \pm 5.01$ | $5.74 \pm 1.44$ | $\mathbf{2.39 \pm 2.43}$ | $\underline{5.44 \pm 1.04}$ |
| | $\Delta_{EO}$ | $10.63 \pm 0.13$ | $9.36 \pm 1.23$ | $5.14 \pm 5.35$ | $3.18 \pm 1.02$ | $\mathbf{1.39 \pm 1.66}$ | $\underline{2.71 \pm 0.58}$ |

## 4. Experiments

### 4.1. Settings

**Datasets.** We conduct experiments on three commonly used datasets(Dong et al., 2022), including German, Bail, and Credit. The statistics of datasets are shown in Table 2.

*Table 2.* Statistic information of three real-world datasets.

| Dataset | German | Bail | Credit |
|---|---|---|---|
| #Nodes | 1,000 | 18,876 | 30,000 |
| #Edges | 22,242 | 321,308 | 1,436,858 |
| #Attr. | 27 | 18 | 13 |
| Sens. | Gender | Race | Age |

**Evaluation Metrics.** We use accuracy and AUC-ROC to evaluate the utility performance of the model, and the stability is measured by the curve of accuracy changes. To evaluate fairness, we use two commonly used fairness metrics, that is, $\Delta DP = |P(\hat{y} = 1|s = 0) - P(\hat{y} = 1|s = 1)|$ and $\Delta EO = |P(\hat{y} = 1|y = 1, s = 0) - P(\hat{y} = 1|y = 1, s = 1)|$. $\hat{y}$ and $y$ denote the prediction of node embedding and label. For $\Delta DP$ and $\Delta EO$, a smaller value indicates a better fairness.

**Baselines.** In our work, we utilize FairVGNN(Wang et al.,

2022), a representative adversarial-based model, as the foundation for our proposed SFG. Subsequently, we compare the accuracy and fairness performance of SFG with several state-of-the-art fair graph learning models. Specifically, we compare the performance of SFG with five baseline methods, i.e., GCN, NIFTY(Agarwal et al., 2021), FairGNN(Dai & Wang, 2021), FairVGNN(Wang et al., 2022), FairSAD(Li et al., 2024). For evaluation of stability, we only compared with adversarial-based model FairVGNN.

**Implementation Details.** We conduct all experiments 5 times and reported average value and variance. For a fair comparison, we tuned the hyperparameters for all methods according to metric $\Delta = ACC + AUC + F1 - \Delta DP - \Delta EO$ on the validation set. For SFG, we use a 2-layer GraphSAGE encoder with hidden dimensions 16 and set the number of generated fair feature views $\mu = 10$ for all datasets. We set the range of the Lipschitz constant $\tau$ is $\{1, 2, 4, 5, 6, 20, 50\}$, which should not be larger than the unconstrained counterpart, for German, Bail, Credit.

### 4.2. Overall Results

We compare the stability of the improved FairVGNN, enhanced with a tighter Lipschitz bound and DRO, against

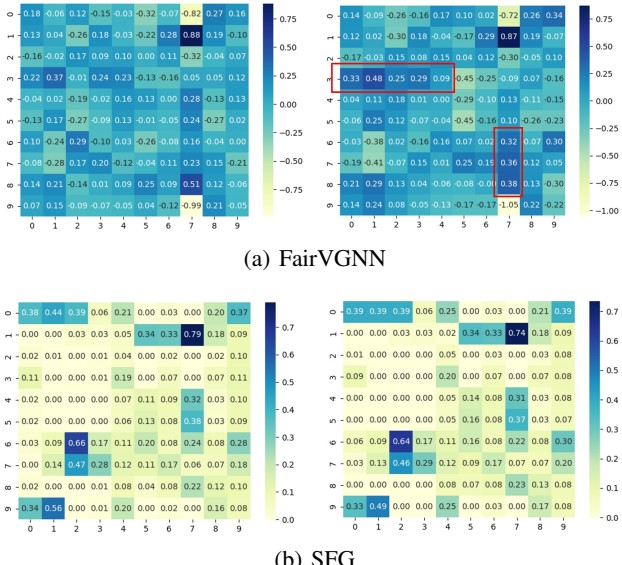

(a) FairVGNN

(b) SFG

*Figure 5.* The weight patch of encoder for FairVGNN and SFG on German dataset.

*Table 3.* Ablation results for performance on German, Bail, and Credit. Reults indicate the DRO module needs to be used together with the Lipschitz constraint module to improve accuracy.

| Dataset | Metric | SFG w/o ct | SFG w/o dro | SFG |
|---|---|---|---|---|
| German | Acc | $69.76 \pm 0.32$ | $70.24 \pm 0.32$ | $70.32 \pm 0.46$ |
| | AUC | $70.36 \pm 2.79$ | $67.67 \pm 3.49$ | $69.38 \pm 4.77$ |
| | $\Delta_{DP}$ | $2.68 \pm 2.89$ | $1.82 \pm 2.73$ | $1.36 \pm 2.61$ |
| | $\Delta_{EO}$ | $2.96 \pm 2.53$ | $1.47 \pm 2.50$ | $0.54 \pm 0.74$ |
| Bail | Acc | $88.76 \pm 0.76$ | $89.06 \pm 1.20$ | $89.49 \pm 1.74$ |
| | AUC | $91.78 \pm 0.59$ | $93.32 \pm 0.67$ | $93.98 \pm 1.41$ |
| | $\Delta_{DP}$ | $3.10 \pm 1.42$ | $3.74 \pm 0.95$ | $3.55 \pm 0.58$ |
| | $\Delta_{EO}$ | $1.50 \pm 0.98$ | $1.27 \pm 0.90$ | $1.11 \pm 0.43$ |
| Credit | Acc | $80.37 \pm 0.05$ | $80.20 \pm 0.23$ | $80.34 \pm 0.11$ |
| | AUC | $74.27 \pm 0.49$ | $73.74 \pm 0.15$ | $73.88 \pm 0.12$ |
| | $\Delta_{DP}$ | $6.71 \pm 0.90$ | $5.48 \pm 0.54$ | $5.44 \pm 1.04$ |
| | $\Delta_{EO}$ | $3.81 \pm 0.65$ | $2.97 \pm 0.52$ | $2.71 \pm 0.58$ |

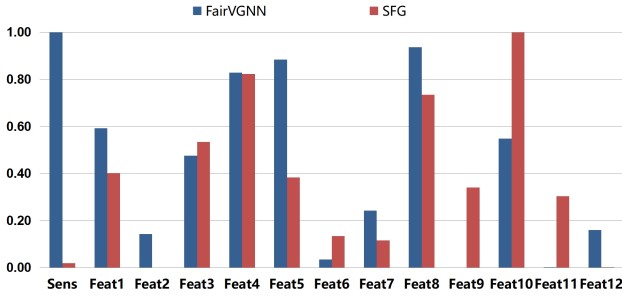

*Figure 6.* The learned mask of the generator on Credit dataset.

its original counterpart. For the utility and fairness of the model, we compare the performance of SFG with four baseline methods and vanilla GCN on the node classification task. As illustrated in Figure 4, the training curve is stable compared to FairVGNN during the convergence stage. Specifically, the amplitude and frequency of fluctuations are better, and the accuracy is generally better than the original model. Table 1 summarizes the results of the SFG comparison against all the baseline methods for real-world datasets. It is evident that SFG outperforms all baseline methods across most evaluation metrics, which indicates the superiority of SFG in achieving better stability and trade-off between model utility and fairness. Compared to FairSAD, our model achieves 5.76% and 2.93% improvement in accuracy for Bail and Credit dataset. FairSAD excels in handling fairness because it utilizes disentangled layers with multiplicatively increasing parameters in the encoder and incorporates an additional neighbor assigner. More experiments results can be found in Appendix D.1.

Notably, SFG not only reduces the frequency of fluctuations, but also significantly reduces the magnitude of fluctuations. In Bail datasets, we can see that the SFG dramatically improved worst accuracy, which may be caused by the generation of a mask that leads to overfitting during fairness training. This enhanced stability and fairness can be attributed to two reasons:(1) By constraining the Lipschitz bound of the generator and encoder during training, we minimize changes of the learned embeddings in trade-off between fairness and utility, thereby enhancing the overall stability of the training process and making it more reliable and trustworthy. (2) Our proposed DRO objective is equiva-

lent to removing the nodes that caused the overfitting, so the stability of the worst-case can be improved. In summary, the experimental results demonstrate the effectiveness of SFG in enhancing stability fairness while preserving trade-off between fairness and utility.

In order to verify whether the Lipschitz bound limits the range of weight changes and leads to the stability of the model, we compared the weight changes of each epoch. As shown in Figure 5, we compared the optimal model weights with the counterpart of the previous epoch, and the German dataset is used. The results indicate that SFG does constrain the fluctuation of weights and can clearly distinguish different features. Specifically, compared to the weight range of [-1, 0.75](FairVGNN) in Figure 5(a), the range of [0, 0.75](Our SFG) in 5(b) exhibits smaller overall fluctuations(43% of FairVGNN). From a micro perspective, the relative weight change(FairVGNN: 7.64%, SFG:1.08%) in the red boxes is calculated as the weight difference between two epochs divided by the total range of variation. This average comparison also reflect that our SFG exhibits smaller fluctuations(14% of FairVGNN) between two epochs. Detailed comparison can be seen in the Appendix D.6.

As shown in Figure 6, we can see that the mask value (probability of keep the channel) of the sensitive attribute (Age)

is close to 0, which means that the sensitive attribute is effectively masked. In addition, most of the features' masks are consistent with the previous models.

### 4.3. Ablation Study

We perform an ablation study to evaluate the impact of each SFG component on enhancing stability and preserving accuracy and fairness. Concretely, we denote SFG w/o ct as removing the Lipschitz bound constraint of the generator and encoder, SFG w/o dro as removing the proposed optimization objective, and SFG w/o ct&dro as removing both of these two modules, i.e., FairVGNN. The training curve of there variants is presented in Figure 7, and the utility and fairness performance are summarized in Table 3.

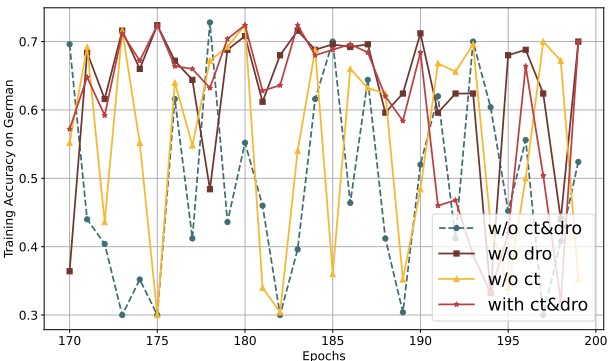

*Figure 7.* Ablation results for stability on German.

We observe that SFG w/o ct is just as bad as the original model in stability, which validates the effectiveness of Lipschitz bound constraint for learning fair node representations stably. Furthermore, SFG with ct&dro perform better than SFG w/o dro, indicating that the proposed DRO module needs to be used with Lipschitz bound constraint, and has played a great role in mitigating extremely unstable anomalies. For fairness performance, SFG with ct&dro is better, except for the German dataset. The accuracy metrics are essentially equivalent.

### 4.4. Parameters Sensitivity Study

The sensitivity of two parameters, i.e. $\tau$ in Equation (9) and $\alpha_{min}$ in Equation (16), was studied mainly. In SFG, the Lipschitz bound $\tau$ controls the contribution of training stability. Specifically, we vary the values of $\tau$ as $\{50, 20, 5, 2\}$ in the German data set and vary the values of $\tau$ as $\{6, 4, 2, 1\}$ in the Bail and Credit datasets. Figure 8 presents the stability results of the parameter $\tau$ analysis of the sensitivity on Credit dataset, the details of the accuracy and fairness performance can be found in the Appendix D.2.

We make the following observation: (1) The SFG remains stable across a wide range of variations in $\tau$, and the overall

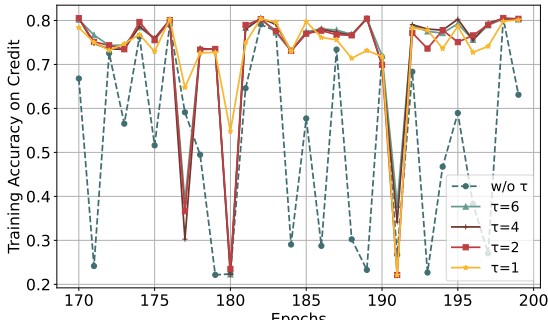

*Figure 8.* Hyperparameter analysis w.r.t. $\tau$ on Credit.

performance gains a slight improvement. (2) When the value of $\tau$ is small, the stability and performance of SFG is worse compared to scenarios where the value of $\tau$ is relatively large.

This observation highlights that Lipschitz bound constraint $\tau$ has a great positive effect on improving the stability of the model and can improve the performance of the model.

Figure 9 presents the stability results of the parameter $\alpha_{min}$ analysis of the sensitivity on Bail dataset, and the accuracy and fairness performance details can be found in the Appendix D.3. The results indicate the dro has significantly improved the stability of the abnormal epoch, which is implemented by removing overfitting nodes.

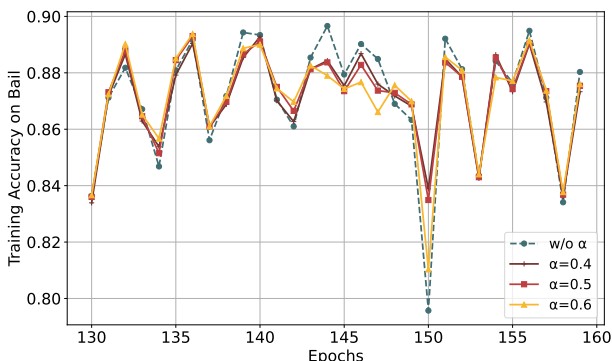

*Figure 9.* Hyperparameter analysis w.r.t. $\alpha_{min}$ on Bail.

### 4.5. Time and Space Complexity

Table 4 outlines the time and space complexity(Derrick et al., 2019; Wu et al., 2020) of the GNN encoder for SFG compared to other baseline fairness methods. We assume $N_i = F$ for layer $i \in \{1, \dots m\}$ and $F \ll K$. $E$ denotes the number of all edges in graph and $n$ represents the number of independent channels in the disentangled layers of FairSAD. We observe that: (1) Compared to FairVGNN, our model introduces only a marginal amount of additional computational and storage overhead during the forward propagation

*Table 4.* Time and space complexity analysis of SFG with baseline fairness methods.

| Model | Forward Time | Forward Space | Backward Time | Backward Space | Parameter Count |
|---|---|---|---|---|---|
| FairVGNN | $mEF + mKF^2$ | $E + mF^2 + mKF$ | $mEF + mKF^2$ | $E + mF^2 + mKF$ | $mF^2$ |
| FairSAD | $n(mEF + mKF^2)$ | $n(E + mF^2 + mKF)$ | $n(mEF + mKF^2)$ | $n(E + mF^2 + mKF)$ | $mnF^2$ |
| SFG | $mEF + mKF^2 + mF^3$ | $E + mF^2 + mKF + mF^2$ | $mEF + mKF^2$ | $E + mF^2 + mKF$ | $mF^2$ |

process, which is attributed to the decoupling of the derived bound from the number of nodes $K$. The increased overhead stems from the complexity associated with performing singular value decomposition (SVD) on the weight matrices. (2) In contrast to FairSAD, we achieve a reduction in the parameter count and computational and storage costs by a factor of the number of independent channels $n$.

## 5. Conclusion

In this paper, we first theoretically derive an easy-to-compute tight upper Lipschitz bound to control the stability of generator and encoder, and use the block coordinate approach which operates in a block-coordinate manner to control weights. Additionally, we use multi-view DRO objective to further enhance the robustness of GNN encoder. Experiments on three real-world datasets demonstrate superior stability and performance of our model.

## Acknowledgements

This research is funded by National Natural Science Foundation of China (No. 62406347).

## Impact Statement

This paper presents work whose goal is to advance the field of Machine Learning. All the work and data are based on existing public datasets and methods, thus there are no potential adverse societal consequences.

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

# A. Related Work

**Fairness in Graph.** Fairness in graph can be mainly divided into two categories, i.e., group fairness(Mandal et al., 2020) and individual fairness(Kang et al., 2020). Group fairness, which emphasizes model's prediction neither favor nor harm certain groups defined by the sensitive attribute, is investigated in this work. As a crucial issue in the field of graph learning, several recent state-of-the-art fair graph models have been proposed to solve fairness. We divide them into two categories: augmentation-based models and adversarial-based models. Augmentation-based models mitigate discrimination through graph augmentation, where sensitive-related information is removed by altering the graph topology or node features, such as NIFTY(Agarwal et al., 2021) and EDITS(Dong et al., 2022). The adversarial-based models force the fairness of node representations by alternatively training the generator and encoder to fool the discriminator and the discriminator to predict the sensitive attributes, and the classifier is used to guarantee the model utility. These models include FairGNN(Dai & Wang, 2021), FatraGNN(Li et al., 2024) and FairVGNN(Wang et al., 2022), which is the main object of our study of stability. Compared to NIFTY, Our model achieves fairness through adversarial training, rather than by learning fair representations via a specific objective function. Additionally, the bound of NIFTY is loose.

**Lipschitz constant.** To the best of our knowledge, there is a relative paucity of literature on the Lipschitz bounds of GNNs, especially for fairness. Pioneering work(Szegedy et al., 2014) points out that the Lipschitz constant of the neural network can be derived as the accumulation of parameters at each layer. Arghal et al. (2022) focused on controlling the Lipschitz constant of GNN filters in the frequency domain, Zhao et al. (2021) proposed a separable Lipschitz bound for GNNs, constraining both the weight and gradient norms of each layer to 1, and Jia et al. (2024) investigated the approximation of the Lipschitz bound via the Jacobian matrix to enhance individual fairness in GNNs from a ranking perspective. There bounds are loose, and have not yet been applied to the stability of graph fairness. Additionally, these bounds are only used to constrain the stability of the encoder and do not take into account other modules associated with the encoder, such as the generator.

# B. Proof of Theorem 1

## B.1. Preliminary Results

Define $L$ as the normalized Laplace matrix, $U$ as the orthogonal matrix after spectral decomposition, where the column vectors are unit eigenvectors and $\Lambda$ as the diagonal matrix of the eigenvalue. In spectral graph theory, graph convolution can be expressed by:

$$y = \sigma(Ug_{(\theta)}U^Tx), \tag{17}$$

where $g_{(\theta)}$ is graph filter with learned parameters, and $\sigma$ is activation function. To represent the above convolution kernel using a $k$-th order polynomial, we have the following expression:

$$
\begin{aligned}
y &= \sigma(U\sum_{k=0}^{K}\theta_k\Lambda^kU^Tx) \\
&= \sigma(\sum_{k=0}^{K}\theta_k(U\Lambda^kU^T)x) \\
&= \sigma(\sum_{k=0}^{K}\theta_k(U\Lambda U^T)^kx) \\
&= \sigma(\sum_{k=0}^{K}\theta_kL^kx).
\end{aligned}
\tag{18}
$$

Expanding to matrix form, we have:

$$Y = \sigma(h(L)XW), \tag{19}$$

where $h(L)$ is a polynomial function of $L$, and $W$ is learned parameter matrix. Let $\sigma$ is Relu, and the node $v$'s embedding at layer $i$ is

$$H_v^{(i)} = Relu(\sum_{u=1}^{K}\varepsilon_{vu}f_u^{(i)}), \tag{20}$$

where $f_u^{(i)} \in \mathbb{R}^{1\times N_i}$ is neighbor node's embedding of $v$ after mapping, and $\varepsilon_{vu}$ is element of $h(L)$.

In the spatial domain, the expression of the graph convolution is Equation (2). Then

$$
\begin{aligned}
H^{(i)} &= Relu(W_0^{(i)} H^{(i-1)} + W_1^{(i)} H^{(i-1)}) \\
&= Relu((Id_K \otimes w_0^{(i)}) H^{(i-1)} + (M \otimes w_1^{(i)}) H^{(i-1)}).
\end{aligned}
\tag{21}
$$

the node $v$'s embedding at layer $i$ is

$$
H_v^{(i)} = Relu(f_v^{(i)} + \sum_{u \in N(v)} \rho_{vu} f_u^{(i)}),
\tag{22}
$$

where $f_v^{(i)}$ is the embedding of node $v$ after mapping, and $\rho_{vu}$ is element of $M$ in Equation (3).

If spectral domain graph convolution is equivalent to spatial domain graph convolution, then we have:

$$
\begin{aligned}
H_v^{(i)} &= Relu(f_v^{(i)} + \sum_{u \in N(v)} \rho_{vu} f_u^{(i)}) \\
&= Relu(\sum_{u \in N(v) \cup \{v\}} \rho_{vu} f_u^{(i)}) \\
&= Relu(\sum_{u=1}^{K} \varepsilon_{vu} f_u^{(i)}).
\end{aligned}
\tag{23}
$$

Equation (18) indicates $h(L)$ contains a self-loop. Expanding to matrix form, we have,

$$
M + I = h(L),
\tag{24}
$$

where the parameters of mapping are assumed to be same. This is only suitable for the GraphSAGE.

For a more general expression, $w_0^{(i)}$ is 0 for every $i \in 1, \dots, m$, and $M$ contains a self-loop, then $M = h(L)$. This is consistent with previous work(Gama et al., 2020) where $L$ can be considered as a graph shift operator(GSO).

According to Liao et al. (2024) and Chen et al. (2023), $h(L)$ can cover low-pass, high-pass, and band-pass graph filters. Taking BernNet as an example,

$$
h(L) = \sum_{k=0}^{K} \theta_k \frac{1}{2^K} \binom{K}{k} (2I - L)^{K-k} L^k.
\tag{25}
$$

When $\theta_{K/2} = 1$ and other $\theta_k = 0$, $h(L) = \frac{1}{2^K} \binom{K}{K/2} (2I - L)^{K/2} L^{K/2}$ is an impulse bandpass filter, and $h(\lambda) \in [0, 1]$ is the element of the diagonal matrix $h(\Lambda)$.

Therefore, we have:

$$
M = \begin{cases} h(L) - I & \text{if model is GraphSAGE,} \\ h(L) & \text{otherwise,} \end{cases}
\tag{26}
$$

and the graph frequency response of the filter is $h(\lambda) - 1$ and $h(\lambda)$ respectively.

### B.2. Proof of Theorem 1

Since $M$ is equivalent to the graph filter $h(L)$, the $M$ is symmetric with non-negative elements. For GraphSAGE, $M = D^{-1/2} A D^{-1/2} = I - L = h(L) - I$, so $h(L) = 2I - L$, and $h(\lambda) - 1 \in (-1, 1)$. This implies that its trace is zero, and it has negative and positive eigenvalues. For the sake of clarity, we use $h(\lambda)$ as the frequency response function of the graph filter. By performing spectral decomposition on M, we obtain the following expression:

$$
M = h(L) = U h(\Lambda) U^T,
\tag{27}
$$

where $h(\Lambda)$ is a diagonal matrix of eigenvalues, and $h(\lambda)$ is a element of the diagonal matrix. For example, $h(\lambda)$ is in range of 0-2 when $h(L) = L$, and it is a high-pass graph filter.

Suppose that M satisfies the above conditions. According to Equation (2) and Equation (3), we have:

$$
\begin{aligned}
W^{(i)} &= W_0^{(i)} + W_1^{(i)} \\
&= Id_K \otimes w_0^{(i)} + h(L) \otimes w_1^{(i)} \\
&= (UU^T) \otimes w_0^{(i)} + (Uh(\Lambda)U^T) \otimes w_1^{(i)} \\
&= (U \otimes Id_{N_i})(Id_K \otimes w_0^{(i)})(U^T \otimes Id_{N_{i-1}}) + (U \otimes Id_{N_i})(h(\Lambda) \otimes w_1^{(i)})(U^T \otimes Id_{N_{i-1}}) \\
&= (U \otimes Id_{N_i})(Id_K \otimes w_0^{(i)} + h(\Lambda) \otimes w_1^{(i)})(U^T \otimes Id_{N_{i-1}}).
\end{aligned}
\tag{28}
$$

The fourth equality above is obtained based on the distributive and associative laws of Kronecker products, i.e., $(A \otimes B)(C \otimes D) = (AC) \otimes (BD)$, and the matrices have compatible dimensions for multiplication. Additionally, we get following expression:

$$
\begin{aligned}
(U \otimes Id_{N_i})(U \otimes Id_{N_i}) &= (UU^T) \otimes (Id_{N_i} Id_{N_i}) \\
&= Id_K \otimes Id_{N_i} \\
&= Id_{KN_i}.
\end{aligned}
\tag{29}
$$

Finally, we get the accumulation of all parameters when the fair graph model is equipped with a generator of mask, it's

$$
W^{(m)} \ldots W^{(1)} W_g = (U \otimes Id_{N_m})(Id_K \otimes w_0^{(m)} + h(\Lambda) \otimes w_1^{(m)}) \ldots (Id_K \otimes w_0^{(1)} + h(\Lambda) \otimes w_1^{(1)})(U^T \otimes Id_{N_0}) W_g,
\tag{30}
$$

where $w_g \in \mathbb{R}^{1 \times N_0}$ is the parameter of generator, and we first form it into a diagonal matrix $W_{g_1} \in \mathbb{R}^{N_0 \times N_0}$. Then we tile it along the diagonal K times and $W_g \in \mathbb{R}^{KN_0 \times KN_0}$ is also a diagonal matrix.

Let us analyze the interchangeability of $(U^T \otimes Id_{N_0})$ and $W_g$. $(U^T \otimes Id_{N_0})$ can be seen as $Id_{N_0}$ being tiled $K \times K$ times. Since the product of two diagonal matrices is commutative, so we have

$$
Id_{N_0} W_{g_1} = W_{g_1} Id_{N_0}.
\tag{31}
$$

For $(U^T \otimes Id_{N_0})W_g$, its each block is $U_{ij}^T Id_{N_0} W_{g_1}$, and $U_{ij}^T$ is the element of $U^T$ at position $(i, j)$. Therefore, $(U^T \otimes Id_{N_0})W_g$ is commutative, i.e.,

$$
(U^T \otimes Id_{N_0})W_g = W_g(U^T \otimes Id_{N_0}).
\tag{32}
$$

times For every $i \in 1, \ldots m$, $Id_K \otimes w_0^{(i)} + h(\Lambda) \otimes w_1^{(i)}$ is a block diagonal matrix of size $(KN_i) \times (KN_{i-1})$ having K block diagonal elements, and Its $k$-th diagonal block is expressed as $w_0^{(i)} + h_k(\lambda)w_1^{(i)}$.

In summary, $k$-th diagonal block of eigenvalue matrix of $W^{(m)} \ldots W^{(1)} W_g$ is expressed as:

$$
\begin{aligned}
&(w_0^{(m)} + h_k(\lambda)w_1^{(m)}) \ldots (w_0^{(1)} + h_k(\lambda)w_1^{(1)})W_{g_1} \\
&= (w_0^{(m)} + h_k(\lambda)w_1^{(m)}) \ldots (w_0^{(1)} + h_k(\lambda)w_1^{(1)}) \odot w_g,
\end{aligned}
\tag{33}
$$

where $\odot$ represents element-wise product supporting broad-cast.

Acording to the Perron-Frobenius theorem, we have

$$
\mid h_k(\lambda) \mid \leq h_{max}(\lambda), (\forall k \in 1, \ldots, K),
\tag{34}
$$

where $h_{max}(\lambda)$ is the maximal frequency response of the graph filter.

So we get the Theorem 1:

$$
\begin{aligned}
Lip(f) &= \| W^{(m)} \ldots W^{(1)} W_g \|_S \\
&= \| (w_0^{(m)} + h_{max}(\lambda)w_1^{(m)}) \ldots (w_0^{(1)} + h_{max}(\lambda)w_1^{(1)}) \odot w_g \|_S .
\end{aligned}
\tag{35}
$$

## C. Proof of Proposition 4

Define $P_k$ as the distribution of the $k$-th domain group, $P_{all}$ is the distribution of all views, $\alpha_i^k$ is the $k$-th group's size of $i$-th view and $\boldsymbol{v}_i$ is the represents distance between the true mask and $i$-th generated mask. We set:

$$P_{all} = \sum_{i=1}^{\mu} \boldsymbol{v}_i \sum_{j=1}^{K} \alpha_i^j P_j. \tag{36}$$

Then, wee have:

$$
\begin{aligned}
D_{\chi^2}(P_k \| P_{all}) &= \int_x \left( \frac{P_k(x)}{P_{all}(X)} - 1 \right)^2 P_{all}(x) dx \\
&= \int_x \left( \frac{P_k(x)}{\sum_{i=1}^{\mu} \boldsymbol{v}_i \sum_{j=1}^{K} \alpha_i^k P_j(x)} - 1 \right)^2 P_{all}(x) dx \\
&\leq \int_x \left( \frac{P_k(x)}{\sum_{i=1}^{\mu} \boldsymbol{v}_i \alpha_i^k P_k(x)} - 1 \right)^2 P_{all}(x) dx \\
&= \int_x \left( \frac{1}{\sum_{i=1}^{\mu} \boldsymbol{v}_i \alpha_i^k} - 1 \right)^2 P_{all}(x) dx.
\end{aligned}
\tag{37}
$$

For simplicity, we assume that $\boldsymbol{v}_i$ follows a uniform distribution, then:

$$
\begin{aligned}
D_{\chi^2}(P_k \| P_{all}) &\leq \int_x \left( \frac{1}{(\alpha^k)_{min}} - 1 \right)^2 P_{all}(x) dx \\
&= r_k,
\end{aligned}
\tag{38}
$$

where $(\alpha^k)_{min}$ is the smallest size of the $k$-th group among all views, and where $r_k := (1/(\alpha^k)_{min} - 1)^2$ is the robustness radius. Therefore, $r_{max} := (1/\alpha_{min} - 1)^2$ where $\alpha_{min}$ is the smallest size of the entire group among all views. It is easy to see that the distribution $\mathbf{v}$ is not uniform, and this conclusion still holds.

We proved $P_k \in B(P_{all}, r_k)$. Since the sup is over all $Q \in B(P_{all}, r_k)$, the upper bound is consistent.

According to Lemma 3.2 and Lemma 3.3, we can get the following proposition.

Let the distribution of the fair feature view generated be denoted by $P_i, i \in 1, \dots, \mu$. Minimizing the worst-case risk on all views, is equal to minimize the following expression:

$$\mathcal{R}_{sfg}(f) := \sum_{i=1}^{\mu} \mathbf{v}_i (C(\mathbb{E}_{P_i}[[l(f(G_v), y_v) - \eta]_+^2])^{1/2} + \eta), \tag{39}$$

where $C = (2(1/\alpha_{min} - 1)^2 + 1)^{1/2}$, and $\alpha_{min}$ is the smallest size of the entire group among all views. $\mathbf{v} \in \Delta_\mu$, $\Delta_\mu$ is a $(\mu - 1)$-dimensional probability simplex, and $\mathbf{v}_i$ represents distance between the true mask and $i$-th generated mask.

## D. Additional Results

### D.1. Effect of Each Component for Stability and Performance

We provide detailed training stability for different components on all datasets.

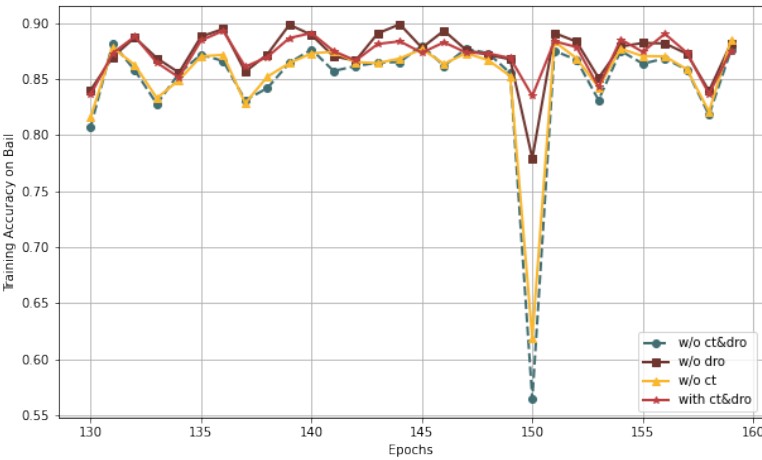

*Figure 10.* Training accuracy curve of the convergence phase on Bail

As shown in Figure 10, we observe that SFG w/o ct is just as bad as the original model in stability for Bail dataset, which validates the effectiveness of Lipschitz bound constraint for learning fair node representations stably. Furthermore, SFG with ct&dro perform better than SFG without dro, indicating that the proposed DRO module needs to be used with the Lipschitz bound constraint and has played an important role in mitigating extremely unstable anomalies.

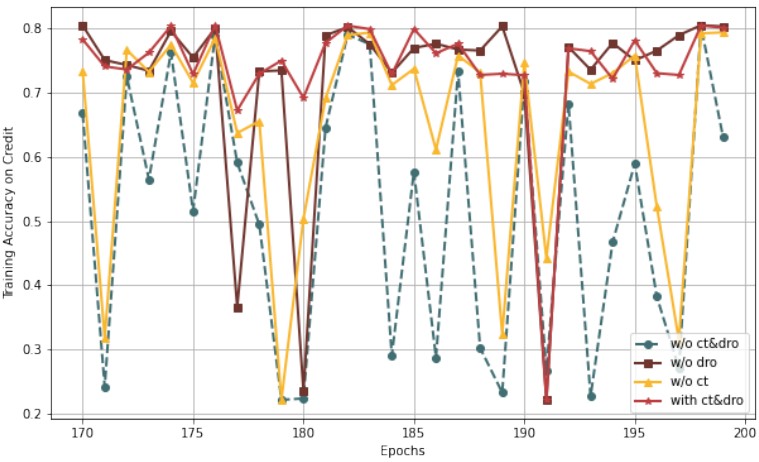

*Figure 11.* Training accuracy curve of the convergence phase on Bail

As shown in Figure 11, We observe that SFG w/o ct is just as bad as the original model in stability for Credit dataset, which validates the effectiveness of Lipschitz bound constraint for learning fair node representations stably. Furthermore, SFG with ct&dro perform better than SFG w/o dro, indicating that the proposed DRO module needs to be used with Lipschitz bound constraint, and it has played a great role in mitigating extremely unstable anomalies.

In summary, the Lipschitz module is to improve the overall stability, and the DRO module is to improve the stability of a few abnormal epochs.

### D.2. Effect of the Lipschitz Constraint $\tau$ for Stability and Performance

We provide detailed training stability and performance for different the Lipschitz constraint $\tau$ on all datasets.

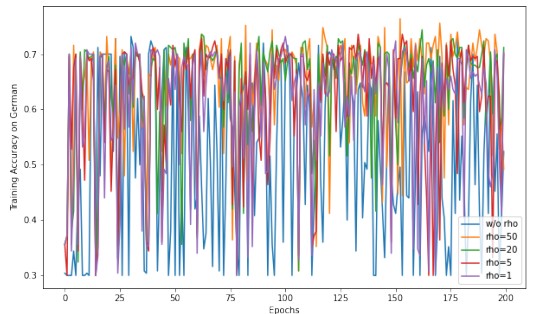

(a) Complete training accuracy curve on German

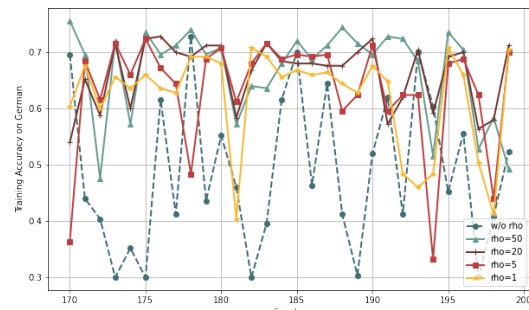

(b) Training accuracy curve of the convergence phase on German

*Figure 12.* Stability of SFG on German. (a) visualizes complete the accuracy curve. (b) visualizes the accuracy curve in the convergence phase.

*Table 5.* Detailed result of parameter $\tau$ sensitivity analysis for performance on German.

| Dataset | Metric | SFG w/o ct | $\tau = 2$ | $\tau = 5$ | $\tau = 20$ | $\tau = 50$ |
|---------|--------|------------|-----------|-----------|------------|------------|
| German | Acc | 70.00 | 70.80 | 70.00 | 70.40 | 69.60 |
|  | AUC | 71.81 | 70.60 | 70.29 | 69.18 | 66.34 |
|  | $\Delta_{DP}$ | 0 | 5.95 | 0 | 3.60 | 1.06 |
|  | $\Delta_{EO}$ | 0 | 3.68 | 0 | 0.21 | 1.79 |

In Figure 12, we plot the complete accuracy curve of FariVGNN and SFG with the Lipschitz constraint against epochs, and the accuracy curve in the convergence phase. SFG with the Lipschitz constraint is significantly more stable than FairVGNN on German dataset, and the best stability is achieved when the $\tau$ is in the range of 20-50. As $\tau$ decreases, the stability gains diminish. As summarized in Table 5, in all ranges of $\tau$, the accuracy is comparable to the original model, and the fairness is slightly reduced.

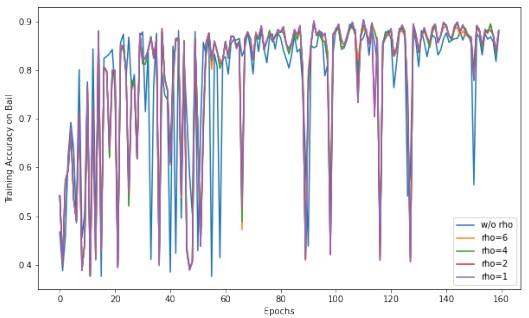

(a) Complete training accuracy curve on Bail

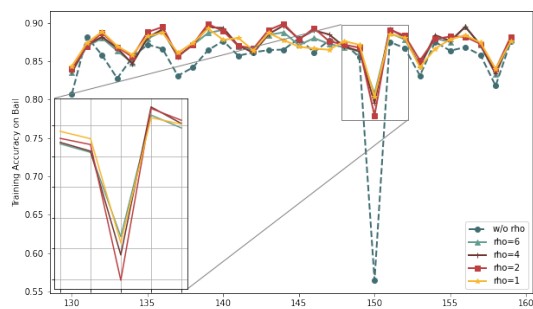

(b) Training accuracy curve of the convergence phase on Bail

*Figure 13.* Stability of SFG on Bail. (a) visualizes complete the accuracy curve. (b) visualizes the accuracy curve in the convergence phase.

*Table 6.* Detailed result of parameter $\tau$ sensitivity analysis for performance on Bail.

| Dataset | Metric | SFG w/o ct | $\tau = 1$ | $\tau = 2$ | $\tau = 4$ | $\tau = 6$ |
|---------|--------|------------|------------|------------|------------|------------|
| Bail | Acc | 89.43 | 89.26 | 90.36 | 89.34 | 89.09 |
| | AUC | 92.23 | 93.94 | 94.21 | 93.46 | 92.81 |
| | $\Delta_{DP}$ | 5.26 | 4.49 | 5.08 | 4.83 | 4.73 |
| | $\Delta_{EO}$ | 4.21 | 2.55 | 1.96 | 1.06 | 1.25 |

In Figure 13, we plot the complete accuracy curve of FariVGNN and SFG with the Lipschitz constraint against epochs, and the accuracy curve in the convergence phase. SFG with the Lipschitz constraint is significantly more stable than FairVGNN on Bail dataset, and the best stability is achieved when the $\tau$ is in the range of 4-6. As $\tau$ changes, the stability is generally unchanged. As summarized in Table 6, in all ranges of $\tau$, both accuracy and fairness have been improved, especially $\Delta eo$.

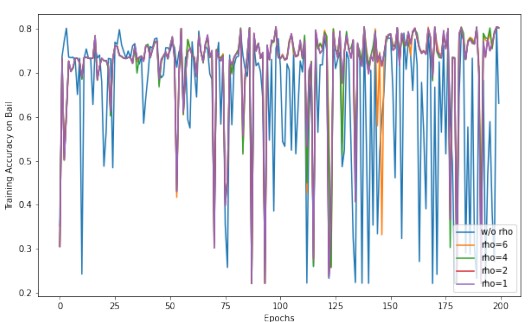

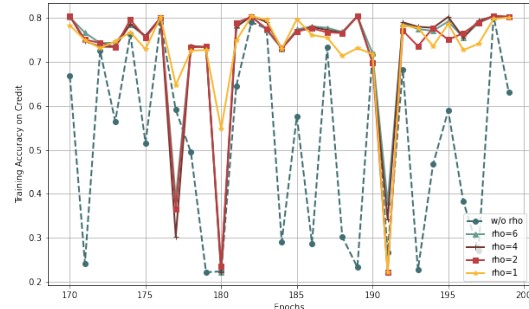

(a) Complete training accuracy curve on Credit    (b) Training accuracy curve of the convergence phase on Credit

*Figure 14.* Stability of SFG on Credit. (a) visualizes complete the accuracy curve. (b) visualizes the accuracy curve in the convergence phase.

*Table 7.* Detailed result of parameter $\tau$ sensitivity analysis for performance on Credit.

| Dataset | Metric | SFG w/o ct | $\tau = 1$ | $\tau = 2$ | $\tau = 4$ | $\tau = 6$ |
|---------|--------|------------|------------|------------|------------|------------|
| Credit | Acc | 80.35 | 80.03 | 80.19 | 79.83 | 80.29 |
| | AUC | 74.37 | 72.09 | 74.07 | 73.85 | 73.59 |
| | $\Delta_{DP}$ | 6.00 | 4.30 | 4.81 | 4.58 | 4.61 |
| | $\Delta_{EO}$ | 3.48 | 1.61 | 2.63 | 1.95 | 1.94 |

In Figure 14, we plot the complete accuracy curve of FariVGNN and SFG with the Lipschitz constraint against epochs, and the accuracy curve in the convergence phase. SFG with the Lipschitz constraint is significantly more stable than FairVGNN on Credit dataset, and the best stability is achieved when the $\tau$ is in the range of 4-6. As $\tau$ changes, the stability is generally unchanged. As summarized in Table 7, in all ranges of $\tau$, both accuracy and fairness have been improved, especially $\Delta eo$.

### D.3. Effect of the DRO Objective $\alpha_{min}$ for Stability and Performance

We provide detailed training stability and performance for different hyper-parameter $\alpha_{min}$ on all datasets.

*Table 8.* Detailed result of parameter $\alpha_{min}$ sensitivity analysis for performance on German.

| Dataset | Metric | SFG with ct | $\alpha_{min} = 0.4$ | $\alpha_{min} = 0.5$ | $\alpha_{min} = 0.6$ |
|---------|--------|-------------|----------------------|----------------------|----------------------|
| German | Acc | 70.00 | 69.19 | 69.60 | 69.60 |
| | AUC | 70.84 | 70.50 | 71.18 | 71.80 |
| | $\Delta_{DP}$ | 0 | 1.26 | 1.06 | 1.06 |
| | $\Delta_{EO}$ | 0 | 1.89 | 1.78 | 1.78 |

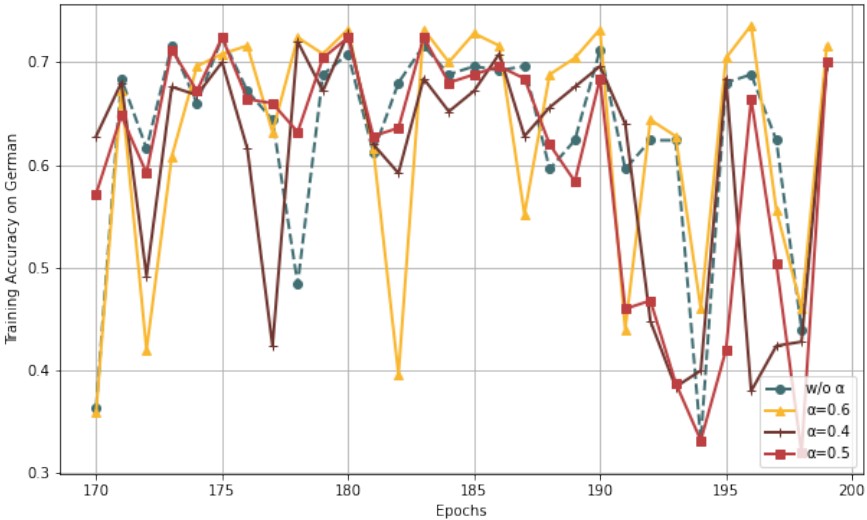

*Figure 15.* Training accuracy curve of the convergence phase on German

In Figure 15, we plot the accuracy curve of SFG with different $\alpha_{min}$ in the convergence phase, and SFG is controlled by the Lipschitz constraint($\tau = 20$). SFG has a positive effect on alleviating the stability of a few abnormal epochs, which may be implemented by dropping some overfitting nodes in chasing fairness. The best stability is achieved when the $\alpha_{min}$ is 0.5, and 0.6 is an inappropriate value and the result is also bad. As summarized in Table 8, the accuracy is comparable to SFG w/o ct, and the fairness is slightly reduced. In summary, the use of DRO objective can alleviate abnormal fluctuations caused by very few epochs and make the model more stable. Notably, the anomalies in the last few epochs are likely due to overfitting, a problem exacerbated by the exceptionally small size of the dataset.

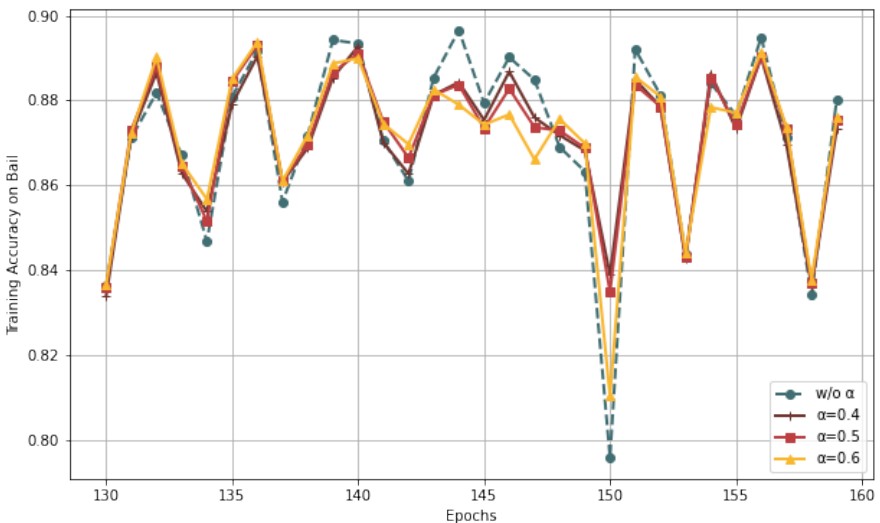

*Figure 16.* Training accuracy curve of the convergence phase on Bail

*Table 9.* Detailed result of parameter $\alpha_{min}$ sensitivity analysis for performance on Bail.

| Dataset | Metric | SFG with ct | $\alpha_{min} = 0.4$ | $\alpha_{min} = 0.5$ | $\alpha_{min} = 0.6$ |
|---------|--------|-------------|---------------------|---------------------|---------------------|
| Bail | Acc | 89.65 | 88.76 | 89.31 | 89.46 |
| | AUC | 93.55 | 94.05 | 94.87 | 93.81 |
| | $\Delta_{DP}$ | 4.49 | 1.26 | 4.20 | 4.73 |
| | $\Delta_{EO}$ | 2.03 | 1.89 | 1.70 | 2.79 |

In Figure 16, we plot the accuracy curve of SFG with different $\alpha_{min}$ in the convergence phase, and SFG is controlled by the Lipschitz constraint($\tau = 4$). SFG has a positive effect on alleviating the stability of a few abnormal epochs, and the best stability is achieved when the $\alpha_{min}$ is in the range of 0.4-0.5. As summarized in Table 9, the accuracy and fairness are comparable to SFG with ct, and even slightly improved when $\alpha_{min}$ is 0.5. In summary, the use of DRO objective can alleviate abnormal fluctuations caused by very few epochs and make the model more stable.

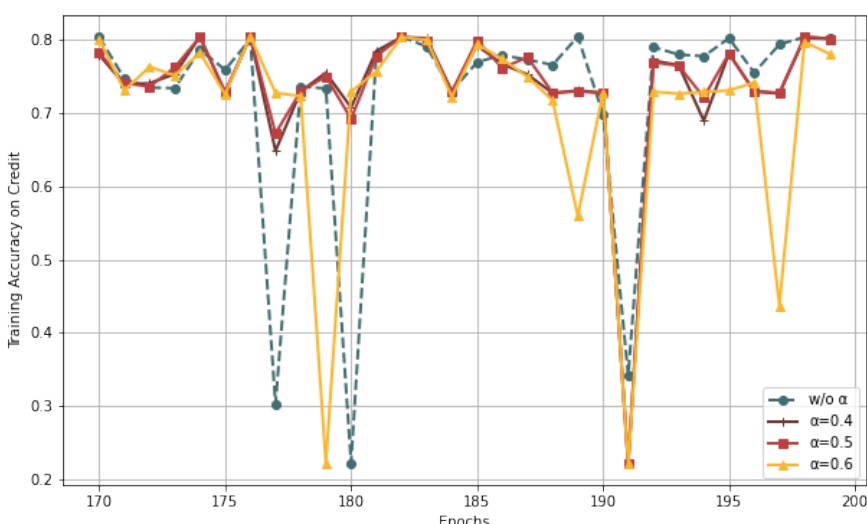

*Figure 17.* Training accuracy curve of the convergence phase on Credit

*Table 10.* Detailed result of parameter $\alpha_{min}$ sensitivity analysis for performance on Credit.

| Dataset | Metric | SFG with ct | $\alpha_{min} = 0.4$ | $\alpha_{min} = 0.5$ | $\alpha_{min} = 0.6$ |
|---------|--------|-------------|---------------------|---------------------|---------------------|
| Credit | Acc | 80.32 | 80.53 | 80.36 | 80.62 |
| | AUC | 73.71 | 73.50 | 73.64 | 73.87 |
| | $\Delta_{DP}$ | 4.76 | 5.85 | 5.40 | 5.10 |
| | $\Delta_{EO}$ | 2.16 | 3.02 | 2.73 | 2.59 |

In Figure 17, we plot the accuracy curve of SFG with different $\alpha_{min}$ in the convergence phase, and SFG is controlled by the Lipschitz constraint($\tau = 2$). SFG has a positive effect on alleviating the stability of a few abnormal epochs, and the best stability is achieved when the $\alpha_{min}$ is in the range of 0.4-0.5. As summarized in Table 10, the accuracy and fairness are comparable to SFG with ct.

## D.4. Accuracy Near the Optimal Model

We provide the accuracy near the optimal model on all datasets.

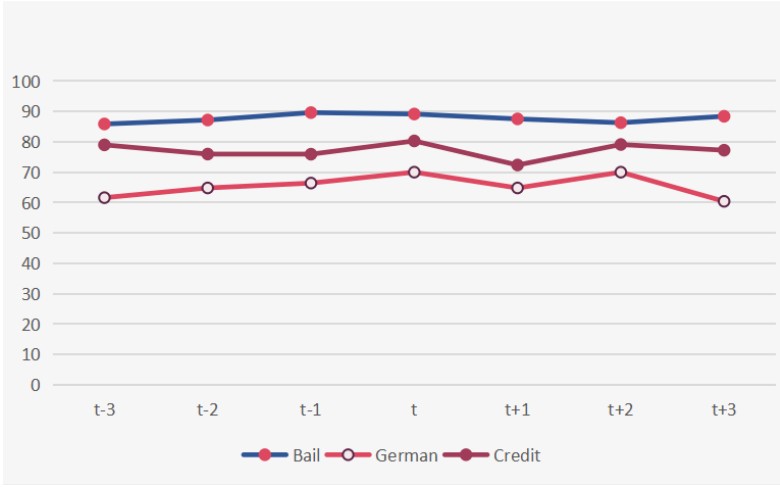

*Figure 18.* Training accuracy near the optimal model

We observe that accuracy near the optimal model is stable in Figure 18, which validates the effectiveness of our proposed model.

### D.5. Model Implementation Details

Fairness-aware view generator aims to mask sensitive features by learning mask weight($g_{\Theta_g}$) for each feature channel, and it has a **learnable sampling score** $s_i$ which denotes the probability of keeping channel $i$. To make the network differentiable, we use the Gumbel-Softmax trick to approximate the categorical Bernoulli distribution. Therefore, the generated mask value is:

$$p_i = \sigma((log(\frac{s_i}{1 - s_i}) + (g_i^1 - g_i^2))/s), \quad \forall i \in 1, \ldots, d, \tag{40}$$

where $g_i^1, g_i^2 \sim Gumbel(0, 1)$, and $\sigma$ is the sigmoid activation, and $s$ is a parameter that controls the smoothness of samples. Then, we could directly multiply the feature channel$X_{:i}$ by the mask weight $p_i$ to get the fair nodes features, i.e.,

$$\tilde{X} = X \odot \mathbf{m}, \tag{41}$$

where $\mathbf{m} = [p_0, p_1, \ldots, p_{d-1}]$ is the learned mask.

In order to learn from both the topology structure and nodes features, we use **2-layer GraphSAGE** as our encoder, i.e., GNN Encoder. In our work, we focus on constraining the weights of GNN Encoder to get a stable node representation. To make encoder robust to different fair views, we employ a multi-view DRO objective to chase fairness.

The Classifier and Discriminator are designed to predict node labels and sensitive attributes, and both consist of a **single-layer MLP** ($d_{\Theta_d}, c_{\Theta_c}$). The objective of these two modules are BCE loss, can be expressed by:

$$L_d = -\mathbb{E}_{v_i \in V}(S_i log(\tilde{S}_i) + (1 - S_i)log(1 - \tilde{S}_i)) \tag{42}$$

and

$$L_c = -\mathbb{E}_{v_i \in V}(y_i log(\tilde{y}_i) + (1 - y_i)log(1 - \tilde{y}_i)), \tag{43}$$

where $y_i$ and $S_i$ denote the labels for nodes and sensitive attributes, respectively. $H_v^{(L)} = e_{\Theta_e}(G_v, \mathbf{m})$ is learned embedding of node $v$ after generator and encoder. $\tilde{S}_i = d_{\Theta_d}(H_i^{(L)})$ is sensitive attribute prediction and $\tilde{y}_i = c_{\Theta_c}(H_i^{(L)})$ is the node label prediction.

The goal of adversarial training is to learn fair view representation by generator and encoder, which masks as much sensitive-relevant information as possible. **This work focuses on optimizing the model using adversarial training while also emphasizing fairness and utility.** In other words, after encoding by the encoder, the stable representations should consider both fairness and utility. The optimization objective of adversarial training can be expressed by:

$$\max_{\Theta_d} L_d = \mathbb{E}_{v_i \in V}(S_i log(\tilde{S}_i) + (1 - S_i)log(1 - \tilde{S}_i)), \tag{44}$$

and

$$\min_{\Theta_g} L_g = \mathbb{E}_{v_i \in V}(\tilde{S}_i - 0.5) + \parallel \mathbf{m} - \mathbf{1}_d \parallel_2^2, \tag{45}$$

where $\mathbf{m} = g_{\Theta_g}(X)$ is the mask weights of all channel generated by generator.

### D.6. Weights Comparison Between FairVGNN and SFG

We list the absolute weight changes between two epochs in following table(both are uniformly rescaled to the range [0, 1]), and the results reflect that our method can indeed limit the fluctuation range of the weights.

*Table 11.* Red box 1 in the weight patch

| Model | FairVGNN | | | | | SFG | | | | |
|---|---|---|---|---|---|---|---|---|---|---|
| epoch $optimal - 1$ | 0.697 | 0.783 | 0.566 | 0.709 | 0.703 | 0.147 | 0.000 | 0.000 | 0.013 | 0.253 |
| epoch $optimal$ | 0.760 | 0.846 | 0.714 | 0.737 | 0.623 | 0.120 | 0.000 | 0.000 | 0.000 | 0.267 |
| absolute change | 0.063 | 0.063 | 0.148 | 0.028 | 0.080 | **0.027** | **0.000** | **0.000** | **0.013** | **0.014** |

*Table 12.* Red box 2 in the weight patch

| Model | FairVGNN | | | SFG | | |
|---|---|---|---|---|---|---|
| epoch $optimal - 1$ | 0.663 | 0.703 | 0.863 | 0.320 | 0.080 | 0.293 |
| epoch $optimal$ | 0.754 | 0.777 | 0.790 | 0.293 | 0.093 | 0.307 |
| absolute change | 0.091 | 0.074 | 0.073 | **0.027** | **0.013** | **0.014** |

### D.7. Baseline with Smaller Learning Rate

We added a baseline with a smaller learning rate(1e-4, FairVGNN), the three nearly unchanged weights in Figure 19 demonstrate that a smaller learning rate can cause the optimization process to get stuck at saddle points, preventing effective learning of weights. Moreover, in the gradient descent algorithm(Su et al., 2022a; 2021a; 2022b; Zheng et al., 2023) $w = w_0 - \eta \Delta w$, large gradients can still lead to significant weight changes such as Figure 20. This is also the reason why we propose the Lipschitz Constraint. Additionally, a smaller learning rate also reduces utility performance(e.g., 4.17% on Credit) of baseline.

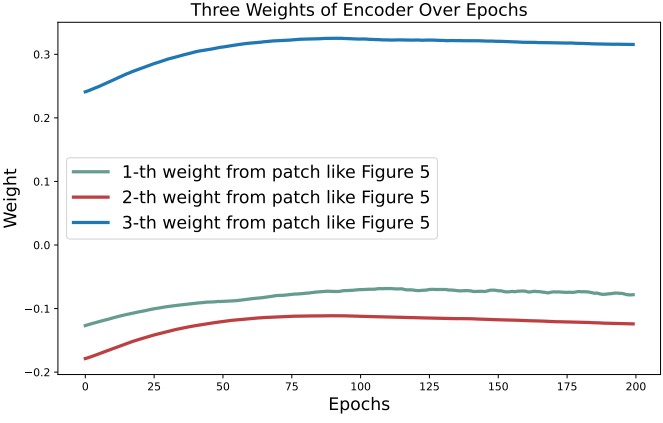

*Figure 19.* Training weights with smaller learning rate

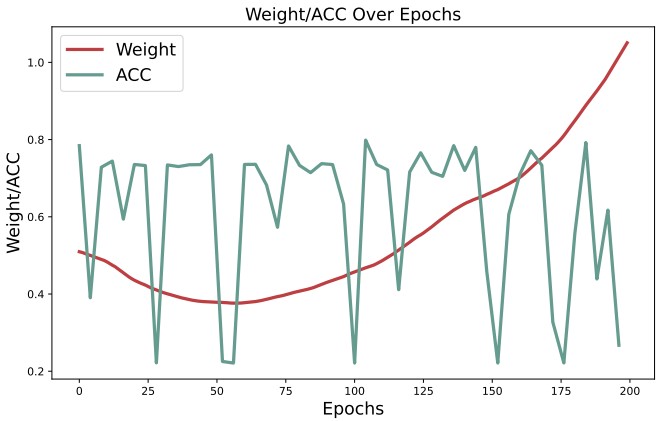

*Figure 20.* Training accuracy/weights with smaller learning rate

### D.8. The Consistent Performance

Table 13 demonstrates that increasing parameters(FairSAD's encoder has four times the parameters of SFG) helps improve performance on small dataset, which fails on large dataset such as Bail and Credit. When FairSAD uses the same number($1\times$) of parameters as SFG on German, SFG outperforms FairSAD by 27% for AUC. The following table also demonstrates that SFG outperforms FairSAD($2\times$) and FairSAD($3\times$), which uses two and three times the number of parameters, by 9.04% and 8.25%, respectively.

*Table 13.* Model Performance with Different Numbers of Parameters

| Model | AUC |
|---|---|
| SFG($1\times$) | $69.38 \pm 4.77$ |
| FairSAD($1\times$) | $54.32 \pm 2.55$ |
| FairSAD($2\times$) | $63.63 \pm 6.99$ |
| FairSAD($3\times$) | $64.09 \pm 3.19$ |

