# OpenReview forum: "Stable Fair Graph Representation Learning with Lipschitz Constraint"
_ICML.cc/2025/Conference — ICML 2025 poster_

### Official Review · Reviewer_skyV · 2025-03-08

**Overall Recommendation:** 3

**Summary:**

This paper proposes a Stable Fair Graph Neural Network (SFG) to address training instability in fairness-aware graph representation learning by introducing a Lipschitz constraint for stability and employing a stochastic optimization algorithm. Extensive experiments demonstrate that SFG outperforms existing methods in both fairness and utility on real-world datasets.

**Claims And Evidence:**

NA

**Essential References Not Discussed:**

NA

**Experimental Designs Or Analyses:**

NA

**Methods And Evaluation Criteria:**

NA

**Other Comments Or Suggestions:**

NA

**Other Strengths And Weaknesses:**

Strengths：
1. The paper is well-written with a clear and logical structure, making it easy to follow.
2. It is grounded in solid theoretical foundations, providing a rigorous basis for the proposed approach.

Weaknesses:
1. The significant accuracy fluctuations observed during the optimization process in Figure 1 leave me puzzled, as they contradict prior research findings. While adversarial learning is known to be inherently unstable, the extent of this instability in the presented results is particularly surprising. Could you clarify the reasons behind this behavior?

**Questions For Authors:**

NA

**Relation To Broader Scientific Literature:**

NA

**Theoretical Claims:**

NA

---

> ### Author Rebuttal · Authors · 2025-03-29
>
> > Q. The reason for significant accuracy fluctuations observed during the optimization process in Figure 1
>
> Thank you for your insightful comment. The significant accuracy fluctuations(even though we have used a small learning rate) observed in Figure 1 are caused by **the addition** of weight fluctuations **in the encoder and mask**, which result from the conflict in optimization objectives. First, adversarial learning would leads to **significant weight fluctuations in the GNN encoder**, such as LECI[1]. Second, current adversarial-based graph models **generate a mask to shield sensitive attributes**, which conflicts with the optimization objective of $\parallel \textbf{m} - \textbf{1}_d \parallel_2^2$, as show in Eq. (45), thereby causing significant fluctuations in the mask weight($w_g$) during the optimization process. We also demonstrate this point through experiment, please refer to the figure(https://anonymous.4open.science/r/SFG-559B/mask.pdf).  The message-passing mechanism of GNN is also a possible reason for the **amplification of weight changes**.  To solve training instability of such graph fair models, we theoretically propose DRO and tight Lipschitz Constraint with mask weight to control weight fluctuations.
>
>
> **References**
> [1] Shurui Gui, Meng Liu, Xiner Li, Youzhi Luo, and Shuiwang Ji. 2023. Joint learning of label and environment causal independence for graph out-of-distribution generalization. In Proceedings of the 37th International Conference on Neural Information Processing Systems (NIPS '23). Curran Associates Inc., Red Hook, NY, USA, Article 174, 3945–3978.

---

### Official Review · Reviewer_v1yW · 2025-03-09

**Overall Recommendation:** 4

**Summary:**

The paper introduces a tight upper bound and distributionally robust optimization to address the challenges of training instability that have been ignored by most previous methods for fair graph representation learning. SFG’s novel upper bound is tight and considers the changes of masks, enhancing training stability while preserving model utility and fairness. The use of a projected stochastic subgradient algorithm makes the non-convex problem convert into a multi-convex problem, facilitating the optimization process.

**Claims And Evidence:**

Yes

**Essential References Not Discussed:**

NA

**Experimental Designs Or Analyses:**

The manuscript describes the experimental settings and designs, and some hyperparameters are detailed. The experimental results and analysis are sufficient. The stability improvement is obvious, and preserving fairness and utility is also improved in most metrics.

**Methods And Evaluation Criteria:**

Yes

**Other Comments Or Suggestions:**

The relationship of $R_{max}$ Eq. (15) and $R_{sfg}$ in Eq. (16) has not been clearly analyzed.

**Other Strengths And Weaknesses:**

Strengths:
The paper proposes a novel tight bound that controls the weights of the model to achieve training stability and innovatively combines the generator with the graph model’s Lipschitz bound for adversarial-based fair GNN. This method addresses the stability-performance (utility and fairness) trade-off in learning fair graph representation, a challenge that has been ignored in prior works. Additionally, the paper proposed Distributionally Robust Optimization to avoid over-fitting when chasing fairness, and it’s a novel way to enhance stability and robustness. The paper clearly articulates the problem being addressed, the proposed solutions, and the derivation process.

Weaknesses:
The update of the weight in Eq (9) has not been explained, and the Multi-convex component in Figure 1 is unclear. Also, the analysis of the GNN complexity is unclear.

**Questions For Authors:**

Please provide an explanation for the weight update in Eq (9) and the Multi-convex component in Figure 1. Please provide the relationship of $R_{max}$ in Eq. (15) and $R_{sfg}$ in Eq. (16). What is the computational method used to calculate the GNN complexity?

**Relation To Broader Scientific Literature:**

The paper analyzes the differences between proposed tight bound and current research. The paper describes the differences between SFG and other models.

**Theoretical Claims:**

Theoretical derivations are detailed and easy to follow.

---

> ### Author Rebuttal · Authors · 2025-03-29
>
> > W1/Q1
> > The update of the weight in Eq (9) has not been explained, and the Multi-convex component in Figure 1 is unclear.
>
> Thank you for your valuable comment. We provide the following clear description for the weight update:
>
> We update the weights layer by layer, and $B_{new}^{(i,t)}$ always uses the updated weights at each step. As shown in the Multi-convex component of Figure 1, the Non-convex optimization is transformed into Multi-convex optimization through a layer-by-layer update approach. The green background indicates that only the i-th layer is updated in the current step, while the light gray indicates that the layer is not updated in the current step. It can also be seen that this approach takes into account the presence of mask weight( $w_g$ \).
>
>
>
> > W2/Q3
> >
> > the analysis of the GNN complexity is unclear.
>
> Thank you for your careful review. We provide the following clear description for the analysis of GNN complexity:
>
> We adopted the approach used by most GNNs[1,2] to calculate the GNN complexity. We analyze the complexity by decomposing MPNN(i.e., Eq. (2)) into three high-level operations:(1) $Z^l=X^lW^l$(feature transformation);(2) $X^{l+1}=AZ^l$(neighborhood aggregation); (3) $\sigma(\cdot)$(activation). We calculate the time and space complexity for each part, and adopt a sparse form based on MPNN. As shown in Table 4, the complexity consists of multiple parts and is detailed.
>
>
>
> > Q3/OCS
> >
> > Please provide the relationship of Rmax in Eq. (15) and Rsfg in Eq. (16).
>
> Thank you for your valuable suggestion. $R_{max}(f)$ denotes the worst-case risk, $R_{sfg}(f)$ is an extension of $F(f, \eta)$ in the multi-view setting. Therefore, optimizing $R_{sfg}(f)$ is equivalent to optimize the loss on the worst-case distribution $R_{max}(f)$, which can improve the robustness and stability across different views.
>
>
> **References**
> [1] Zonghan Wu, Shirui Pan, Fengwen Chen, Guodong Long, Chengqi Zhang, and S. Yu Philip. 2020. A comprehensive survey on graph neural networks. IEEE Transactions on Neural Networks and Learning Systems 32, 1 (2020), 4–24.
>
> [2] Blakely Derrick, Lanchantin Jack, and Qi Yanjun. 2019. Time and Space Complexity of Graph Convolutional Networks. (2019). https://api.semanticscholar.org/CorpusID:269411067

---

### Official Review · Reviewer_BuAy · 2025-03-11

**Overall Recommendation:** 5

**Summary:**

This paper focuses on addressing the challenge of training instability in adversarial-based fair GNN models. To mitigate this issue, it establishes a tight upper Lipschitz bound to regulate stability and leverages Distributionally Robust Optimization (DRO) to improve the encoder’s robustness across different fairness views. The key novelty lies in deriving a precise upper Lipschitz bound for a fair graph model with a generator and incorporating DRO to prevent fairness overfitting, ultimately ensuring stable training.

**Claims And Evidence:**

The claims presented in this paper are generally clear and straightforward. Additionally, the proof process is structured well and effectively communicated.

**Essential References Not Discussed:**

none.

**Experimental Designs Or Analyses:**

I reviewed the descriptions provided in the experimental settings, including the selection of the Lipschitz constant and the model’s key hyperparameters. The paper presents these details clearly, and the chosen hyperparameters are commonly used. In the experimental analysis, the paper examines the improvement margins in stability and utility and also analyzes a specific case, which I find to be a reasonable approach.

**Methods And Evaluation Criteria:**

I believe that enforcing a Lipschitz constraint is essential for ensuring the stability of fair GNNs, and deriving a tight upper bound in the presence of a generator is a novel contribution. Moreover, applying DRO offers an innovative approach to improving the stability of fair models. The datasets utilized in this paper are widely recognized within the fair graph research community. Regarding evaluation criteria, the paper assesses performance from the perspectives of stability, utility, and fairness, which I find to be a reasonable approach.

**Other Comments Or Suggestions:**

The Lipschitz constant range should remain consistent throughout the paper. In Section 4.4, the constant range for the German dataset should be integrated into the experimental settings outlined in Section 4.1.

**Other Strengths And Weaknesses:**

Strengths:

This paper is the first to recognize the instability issue in adversarial-based fair GNN models and introduces a tight Lipschitz bound to enhance stability. The derivation of the upper Lipschitz bound is both theoretically sound and practically justified, with a clear and well-structured explanation. SFG significantly improves stability while maintaining accuracy and fairness in fair graph models. Additionally, this work presents the first empirical study on applying DRO to graph fairness, strengthening the encoder’s robustness across different fair views. A thorough complexity comparison is also provided.

Weaknesses:

The training process could be explained more clearly. The meaning of the circles between Step 3 and Step 4 in Figure 1 is not sufficiently detailed.

**Questions For Authors:**

Is Step 4 of SFG trained alternately or in a unified manner?

Do the circles between Step 3 and Step 4 represent an expansion of the potential distribution range to better approximate the true distribution?

**Relation To Broader Scientific Literature:**

This paper addresses the challenge of training stability in fair GNNs, a critical yet often overlooked issue in the fair graph research community. In the introduction and related work sections, the authors compare SFG with other fairness models, including those involving Lipschitz bounds, providing a well-reasoned discussion. Moreover, these concerns are relevant to a broader range of graph learning tasks.

**Theoretical Claims:**

I have reviewed the paper’s key theoretical derivations, including Theorem 3.1 and Proposition 3.4, and did not find any issues with them.

---

> ### Author Rebuttal · Authors · 2025-03-29
>
> > W1/Q1
> > The training process could be explained more clearly.
>
> Thank you for your valuable suggestion. We provide the following clear description of the training process:
>
> We train SFG in an alternating manner throughout the entire process. In each epoch, we first train discriminator(i.e., $L_d$ in Eq. 42) to recognize the sensitive attributes by generated mask, then we train encoder and classifier using BCE loss(i.e., $L_c$ in Eq. 43). Finally, we train generator(i.e., $L_g$ in Eq. 45) and encoder to achieve fairness. This process is repeated for all epochs. Therefore, SFG used Lipschitz Constraint with mask weights and DRO can enhance stability.
>
>
>
> > W1/Q2
> >
> > The meaning of the circles between Step 3 and Step 4 in Figure 1 is not sufficiently detailed.
>
> Thank you for your insightful comment. Our purpose of using DRO is to narrow the gap(i.e., Shrink Distribution Discrepancy) with the real data distribution by finding the worst-case distribution of the training data, thereby improving the robustness and stability with the mask weight. The circles between Step 3 and Step 4 in Figure 1 **indeed represent an expansion of the potential distribution range to better approximate the real data distribution**.
>
>
>
> > OCS
> >
> > The Lipschitz constant range should remain consistent throughout the paper
>
> Thank you for your careful review. We update the range of constant $\tau$ as {1,2,4,5,6,20,50}.

---

### Official Review · Reviewer_rmL1 · 2025-03-14

**Overall Recommendation:** 3

**Summary:**

This paper presents a novel approach "Stable Fair Graph Neural Network (SFG)" that addresses the issue of instability in adversarial-based fair graph representation learning. The main contributions and findings are as follows (1) The authors derive a tight, easy-to-compute upper Lipschitz bound for the composite model that includes both the sensitive attribute mask generator and the GNN encoder. (2) the paper introduces a stochastic projected subgradient algorithm that employs a block-coordinate update mechanism.

**Claims And Evidence:**

The tight upper Lipschitz bound, which ensures the stability of the framework, is well supported by the developed theorems (Proposition 3.4). However, the claim that SFG effectively constrains weight fluctuations is not convincingly demonstrated in Figure 5. It is difficult to determine which case exhibits lower fluctuation since subfigures (a) and (b) use different legend scales. Additionally, using the absolute change in weights as an indicator of fluctuation would be a more reasonable approach.

**Essential References Not Discussed:**

None

**Experimental Designs Or Analyses:**

This paper claims that Lipschitz bound limits the range of weight changes and leads to the stability of the model. Then, it might be important to include an important baseline: training the model with a smaller learning rate that could also lead to small weight changes.

**Methods And Evaluation Criteria:**

The chosen dataset and metric are appropriate for the evaluation. However, the use of weight changes in Figure5 to reflect the function may not be entirely suitable instead the absolute of weight changes would be more suitable.

**Other Comments Or Suggestions:**

N/A

**Other Strengths And Weaknesses:**

Strengths:
(1) This paper developed a theorem to justify the method.

(2) The paper is generally well written.

Weakness:

(1) The Ablations in Figure 3 show that SFG did not consistently achieve the best performance in AUC such as in German and Credit datasets.

(2) The main results in Table 1 show that SFG can not consistently achieve the best performance across the datasets and the improvements is not significant as well.

**Questions For Authors:**

(1) what is the architecture of the GNN used for SFG? Can this method also work well for other Graph neural networks? such as graphsage or GAT?

(2) In table 1, the proposed SFG did not constantly achieve the best performance across the dataset, what 's the reason for why it performs worse on German dataset?

**Relation To Broader Scientific Literature:**

https://arxiv.org/pdf/2005.02929

**Theoretical Claims:**

Didnt comprehensively evaluate the correctness of the proofs.

---

> ### Author Rebuttal · Authors · 2025-03-29
>
> Thank you for your valuable suggestion and insightful comment.
> > CE. The indicator of weight fluctuation
> > the absolute of weight changes would be more suitable.
>
> We list **the absolute weight changes(FairVGNN: 0.079, SFG:0.018)** between two epochs in following table(both are uniformly rescaled to the range [0, 1]), and it will be supplemented in the revised version to facilitate readers' understanding. Figure 5 also illustrates from both macro and micro perspectives that SFG can constrain weight fluctuations. From a macro perspective, compared to the weight range of **[-1, 0.75]**(FairVGNN) in Figure 5(a), the range of **[0, 0.75]**(Our SFG) in 5(b) exhibits **smaller overall fluctuations(43% of FairVGNN)**.  From a micro perspective, the relative weight change(FairVGNN: **7.64%**, SFG:**1.08%**) in the red boxes is calculated as the weight difference between two epochs **divided by the total range** of variation. This average comparison also reflect that our SFG **exhibits smaller fluctuations(14% of FairVGNN) between two epochs**.
> * The weight changes in red box 2
>
> $$
> \\begin{array}{|c|c|c|}
> \\hline
> \\text{Model} & \\text{FairVGNN} & \\text{SFG} \\\\ \\hline
> \\text{epoch 119} & 0.663, 0.703, 0.863 & 0.320, 0.080, 0.293 \\\\ \\hline
> \\text{epoch 120} & 0.754, 0.777, 0.790 & 0.293, 0.093, 0.307 \\\\ \\hline
> \\text{absolute change} & 0.091, 0.074, 0.073 & \\mathbf{0.027}, \\mathbf{0.013}, \\mathbf{0.014} \\\\ \\hline
> \\end{array}
> $$
>
>
>
> > ED. Baseline with smaller learning rate
>
> We added a baseline with a smaller learning rate(**1e-4**, FairVGNN), the three **nearly unchanged weights** in the figure(https://anonymous.4open.science/r/SFG-559B/small-lr.pdf) demonstrate that **a smaller learning rate can cause the optimization process to get stuck at saddle points**, preventing effective learning of weights. Moreover, in the gradient descent algorithm $w=w_0-\eta \Delta w$ , **large gradients can still lead to significant weight changes**(Please refer to figure https://anonymous.4open.science/r/SFG-559B/small-lr-2.pdf). This is also the reason why we propose the Lipschitz Constraint. Additionally, a smaller learning rate also reduces utility performance(e.g., **4.17% on Credit**) of baseline. Thank you for your suggestion enriched our discussion on training stability, and would be incorporated into the revised version.
>
>
> > W1. The Ablations Performance
> > The Ablations in Table 3 show that SFG did not consistently achieve the best performance in AUC.
>
> Compared to our ablation model **SFG w/o ct**, SFG is the result of a comprehensive consideration of stability, utility, and fairness. As shown in  Figure 7, SFG significantly outperforms SFG w/o ct in terms of training stability, which can inspire further research for training stability of adversarial-based fair models, leading to more reliable and trustworthy models.
>
>
> > Q1. The architecture of the GNN
>
> As elaborated in line 180 and 1080, the architecture of the GNN is **GraphSAGE** in our implementation. The result using **GCN** as backbone in follwing table also proved the effectiveness of our method(SFG outperforms the baseline by **118%** on Bail and 90% **on** Credit for $\Delta EO$). Our tight upper bound(Eq. 5) of SFG is derived based on **the general message-passing framework**(Eq. 1) and is therefore applicable to most GNN backbones, such as GCN, GraphSAGE. Additionally, since the calculation of GAT coefficients involves node representations, it is not applicable to our bound.
>
> $$
> \\begin{array}{|c|c|c|c|c|c|}
> \\hline
> \\text{Dataset} & \\text{Model} & \\text{Acc} & \\text{AUC} & \\Delta_{DP} & \\Delta_{EO} \\\\ \\hline
> \\text{Bail} & \\text{FairVGNN} & 84.76 & 85.62 & 6.45 & 4.89 \\\\ \\hline
> \\text{Bail} & \\text{SFG} & 86.43 & 85.93 & 4.98 & 2.24 \\\\ \\hline
> \\text{Credit} & \\text{FairVGNN} & 78.06 & 71.36 & 6.21 & 4.67 \\\\ \\hline
> \\text{Credit} & \\text{SFG} & 79.74 & 72.06 & 4.54 & 2.45 \\\\ \\hline
> \\end{array}
> \$$
>
>
> > W2/Q2.  The Consistent Performance
> > In table 1, the proposed SFG did not constantly achieve the best performance across the dataset, what 's the reason for why it performs worse on German dataset?
>
> The reason is that increasing parameters(FairSAG's encoder has **four times** the parameters of SFG) helps improve performance on **small dataset**, which fails on **large dataset** such as Bail and Credit. When FairSAD uses the same number(1$\times$) of parameters as SFG on German, SFG outperforms FairSAD by 27% for AUC. The following table also demonstrates that SFG outperforms FairSAD(2$\times$) and FairSAD(3$\times$), which uses two and three times the number of parameters, by 9.04% and 8.25%, respectively.
>
> $$
> \\begin{array}{|c|c|}
> \\hline
> \\text{Model} & \\text{AUC} \\\\ \\hline
> \\text{SFG}(1\\times) & 69.38 \\pm 4.77 \\\\ \\hline
> \\text{FairSAD}(1\\times) & 54.32 \\pm 2.55 \\\\ \\hline
> \\text{FairSAD}(2\\times) & 63.63 \\pm 6.99 \\\\ \\hline
> \\text{FairSAD}(3\\times) & 64.09 \\pm 3.19 \\\\ \\hline
> \\end{array}
> $$

---

> > ### Comment · Reviewer_rmL1 · 2025-04-02
> >
> > I thank the authors for their comprehensive and detailed responses to my questions. After reading their rebuttal, I believe the authors have addressed all the questions I raised. Therefore, i raise  the recommendation to 3.

---

> > > ### Author Response · Authors · 2025-04-02
> > >
> > > We sincerely appreciate your thoughtful consideration of our rebuttals. Your valuable suggestions and insightful comments have deepened our understanding for training stability of adversarial-based fair graph models, which significantly enhances the necessity and importance of our theoretical framework. We are truly grateful for your engagement and for recognizing our work's contributions!

---

### Decision · Program_Chairs · 2025-05-01

**Decision:**

Accept (poster)

**Comment:**

The paper presents a novel approach, Stable Fair Graph Neural Network (SFG), addressing training instability in adversarial-based fair graph representation learning. Key contributions include the derivation of a tight upper Lipschitz bound ensuring stability, and the integration of DRO to enhance model robustness across fairness perspectives.

All reviewers highlight the paper's solid theoretical foundation, particularly appreciating the clearly articulated and rigorous derivations of the Lipschitz bound and the innovative application of DRO. Experimental results convincingly demonstrate improved stability, fairness, and utility over baseline methods.

While some reviewers pointed out minor weaknesses—such as inconsistent performance across certain datasets and initial clarity issues regarding the training procedure—the authors provided thorough, detailed rebuttals addressing all concerns effectively, notably clarifying training procedures, algorithmic components, and complexity analyses.

Overall, reviewers agree the contributions are substantial and novel, and the concerns raised were comprehensively resolved in the rebuttal. Therefore, I recommend acceptance of this submission.